

# Depth-dependence of soil organic carbon additional storage capacity in different soil types by the 2050 target for carbon neutrality

Chirol, Clémentine[1,2], Séré, Geoffroy[1], Redon, Paul-Olivier[3], Chenu, Claire[2], Derrien, Delphine[4]

[1]Université de Lorraine, INRAE, LSE, F-54000 Nancy, France
[2]INRAE, AgroParisTech, Ecologie fonctionnelle et écotoxicologie des agroécosystèmes, Palaiseau, France
[3]Andra, Direction Scientifique & Technologique, Centre de Meuse/Haute-Marne, 55290 Bure, France
[4]INRAE, Centre de Nancy, Biogéochimie des Ecosystèmes Forestiers, F-54280 Champenoux, France

*Correspondence to*: Clémentine Chirol (clementine.chirol@inrae.fr)

**Abstract.** Land planning projects aiming to maximise soil organic carbon (SOC) stocks are increasing in number and scope. In response, a rising number of studies assess SOC additional storage capacities over regional to global spatial scales. In order to provide realistic values transferrable beyond the scientific community, SOC storage capacity assessments should consider the timescales over which this capacity might be reached, considering the effects of C inputs, soil type and depth on soil C dynamics.

This research was conducted in a 320 km$^2$ territory in North-eastern France where eight contrasted soil types have been identified, characterized and mapped thanks to a high density of fully-described soil profiles. Continuous profiles of SOC stocks were interpolated for each soil type and land use (cropland, grassland or forest). Depth-dependent estimates of maximum SOC additional storage capacity using the Hassink equation and a data-driven approach were compared. We used a novel method that uses the data-driven approach to constrain C inputs in a simple model of depth-dependent C dynamics to simulate SOC accrual over 25 years, and mapped the SOC stocks, maximum additional storage capacity and stock evolution.

SOC stocks and maximum additional storage capacities are highly heterogenous over the region of study. Median SOC stocks range from 78 - 333 tC ha$^{-1}$. Data-driven maximum SOC additional storage capacities vary from 19 tC ha$^{-1}$ in forested Leptosols to 197 tC ha$^{-1}$ in grassland Gleysols. Estimations of SOC maximum additional storage capacities based on the Hassink approach led to unrealistic vertical distributions of SOC stock, with particular overestimation in the deeper layers. Crucially, the simulated SOC accrual over 25 years was five times lower than the maximum SOC additional storage capacity (0.57 and 2.5 MgC respectively). Further consideration of depth-dependent SOC dynamics in different soil types is therefore needed to provide targets of SOC storage over timescales relevant to public policies aiming to approach carbon neutrality by 2050.





## 1 Introduction

Soils constitute a carbon reservoir that can help us mitigate for climate change, or conversely accelerate GHG emissions if not managed properly. Objectives for carbon neutrality by 2050 rely on an increase in soil organic carbon (SOC) via changes in land management practices over the coming decades, while preserving existing stocks (Minasny *et al.*, 2017). However, soils are heterogenous and dynamic systems: soil carbon stocks are constantly being mineralized and renewed by new inputs. The spatial heterogeneity of soil carbon stocks and fluxes presents a challenge to soil carbon sequestration strategies. Certain soils may represent large stocks that need to be preserved, while others may have a greater capacity for additional storage.

Historically, the clay and fine silt contents have been used as the main proxy of maximum soil organic carbon storage capacity in the fine fraction (Hassink 1997, Angers *et al.*, 2011). The implication is that once saturation of the total available mineral surface area is reached, soil organic matter can only be stored as particular organic matter (POM), which is considered on average more labile than the mineral-associated organic matter (MAOM) fraction (1-50 years for POM and 10-1000 years for





MAOM, Cotrufo *et al.* (2022)). However, there are arguments that the POM fraction should be included in strategies to preserve and replenish SOC stocks, as it reacts rapidly to changes in land management and OM inputs (Rocci *et al.*, 2021;

Derrien *et al.*, 2023). The aspirational targets of SOC accrual carried by the 4p1000 initiative notably consider both the POM and MAOM fractions (Chen *et al.*, 2019a). Consequently, recent methods have estimated target carbon stocks by the upper percentiles of the total carbon content in a large dataset, estimating it to be representative of the maximum carbon content that can be reached realistically under a specific pedoclimatic condition and under the management practices captured by the dataset (Lal 2016, Chen *et al.*, 2019a).

Previous comparisons between the Hassink method and this data-driven approach in the 0-30 cm and 30-50 cm layers of the soil found similar patterns in their mapping results (Chen *et al.*, 2018, 2019b). However, in arable subsoils, the theoretical saturation in the fine fraction derived from the Hassink equation was found to exceed the maximum SOC additional storage capacities (including both POM and MAOM) derived from the data-driven approach (Chen *et al.*, 2019b). Improved understanding of this discrepancy could be achieved by comparing them explicitly over the whole soil profile.

The importance of estimating SOC stocks over the whole soil profile is increasingly recognised, as the SOC below 20 cm can account for more than 50% of the total stock (Jobbágy & Jackson, 2000; De Vos *et al.*, 2015). Deeply stored carbon can be expected to contribute to SOC stocks over long timescales, since mean carbon age increases with depth from a few years to several hundred years in the first meter (Balesdent *et al.*, 2018). Impacts of management practices on SOC dynamics have been found to vary above and below 30 cm, so the consideration of the whole soil profile is important to provide accurate

recommendations to stakeholders (Tautges *et al.*, 2019). However, due to the difficulty in sampling SOC in deeper horizons, deep carbon dynamics are poorly known compared to shallow soil carbon (Gross & Harrison, 2019): only 11% of the studies included in the metanalysis by Don *et al.* (2023) on C sequestration in soil considered the soil horizons below 30 cm.

The estimation of SOC additional storage capacity needs to be evaluated over timescales relevant to stakeholders, keeping in mind in particular the carbon neutrality objective by 2050. Getting the kinetics of SOC accrual necessitates a model-driven

approach and scenarios of C inputs to the soil (Barré *et al.*, 2017). Mechanistic models of SOC dynamics such as Millenial (Abramoff *et al.*, 2022) are one option to incorporate the effect of climate change and modifications in management practice, but necessitate a lot of input data, therefore simpler models remain valuable to explore (Derrien *et al.*, 2023, Schimel 2023).

Finally, while SOC stocks are strongly affected by land use (Guo & Gifford, 2002), the depth-dependent physico-chemical properties of the soil also play an important role on SOC accumulation and residence time (Kögel-Knabner *et al.*, 2021). Soil

properties that affect SOC stabilization notably include the clay content and exchangeable cations (Rasmussen *et al.*, 2018). High $Ca^{2+}$ concentrations in soils were found to intensify SOC accumulation either through increased occlusion within aggregates or through enhanced SOC association with minerals (Rowley *et al.*, 2021). Low pH values also hinder microbial activity and organic matter degradation, leading to an increased residence time of SOC in the soil (Malik *et al.*, 2018). Soil maps therefore constitute an important tool to spatially assess SOC stocks and fluxes (Wiesmeier *et al.*, 2015).

The main objective of this paper is to explore the effect of soil type on whole-profile SOC dynamics over timescales relevant to stakeholders (25 years), and how SOC accrual over decadal timescales might differ from the maximum SOC additional



storage capacity as estimated by current methods. We have chosen to use a model that has been calibrated over the soil profile by Balesdent *et al.* (2018) using C isotopes tracing over timescales of several decades to several centuries. The originality of our approach resides in the use of data-driven estimates of maximum SOC additional storage capacity combined with the

depth-resolved model by Balesdent *et al.* (2018) to obtain the input scenarios required to reach the maximum SOC stocks of the studied area at the steady state. The combination of pre-existing methods (interpolation of continuous SOC profiles, percentile approach to obtain a continuous profile of maximum SOC stocks, simple model of C dynamics robust at decadal timescales, mapping of SOC accrual scenario after 25 years) is an innovative way of generating realistic results that are transferrable beyond the scientific community.

In this paper, the SOC stock and maximum additional storage capacity per soil type and land cover are represented as continuous profiles as a function of depth, then mapped over a rural region combining croplands, forests and grasslands. Two methods for estimating maximum SOC additional storage capacity, i.e. a classical approach using the carbon saturation method (Hassink, 1997), and a data-driven approach using the upper percentiles of SOC stocks per soil type (Chen *et al.*, 2019a) are compared. Finally, the evolution of SOC stocks under simplified scenarios of yearly incorporated inputs is explored by a

whole-profile SOC dynamic model to produce prospective maps of future stocks after 25 years. The impact of rising temperatures on SOC accrual through an increase in carbon mineralization rates is also considered. We use as case study a region with a large dataset over a small area and where land use change has seen very little variation for 200 years, using a data-driven approach to constrain the C inputs in the SOC dynamics model. We will show how the methodology applied to this data-rich region could later be applied to other zones based on pedological expertise.

## 2 Materials and Methods

### 2.1 Study site and data acquisition

The Perennial Observatory of the Environment (OPE in French) is monitoring since 2007 a 320 km$^2$ area located in the North-Eastern part of France (in Meuse and Haute Marne counties). This observatory operated by the Radioactive Waste Management Agency (ANDRA) aims to follow the environmental impacts of a planned deep underground nuclear waste storage facility. In

the framework of the monitoring program, various environmental data including soil characterization and mapping have been collected.

The OPE study area is dominated by agricultural and forest lands: 55% of the region is occupied by agricultural lands managed by conventional agriculture practices; 29% is occupied by forests dominated by deciduous trees (oak, charm, beech); 14% is occupied by grassland, and less than 2% by urban areas. The region's continental climate is softened by some oceanic

influences. According to data collected by the OPE weather stations from 2009 to 2019, the mean annual temperature is 10.4 °C (+/- 6.2 °C between summer and winter), annual cumulated rainfall is 983 mm (+/- 113) and ETP = 661 mm (+/- 79).

This study uses a total of 198 soil profiles (932 data points) to estimate SOC stocks and additional storage capacity. 86 of these soil profiles were collected within the region of study between 1995 and 2019, and were used along with a 1/50,000 pedological



map (Party *et al.* / Sol Conseil 2019) to classify the soils into eight dominant soil types and define the physico-chemical characteristics of each of their horizons, such as pH, CaCO3, texture and rock fragment content (See measurement protocols in Appendix Table 1).

The eight identified soil types can be broadly divided based on the geological parent materials and the geomorphology of the region (Figure 1). On the plateaus, preserved detritic Cretaceous layers from the Valanginian stage with high concentrations of silt and sand lead to the formation of Eutric and Dystric Cambisols, with locally Podzosols reaching deeper than 2m. On the hillslopes and in the valleys, the parent materials are Tithonian limestones and Kimmeridgian marls and limestones, leading to the formation of Calcaric to Hypereutric Cambisols with high rock fragment contents in the deeper horizons. Soils on the hillslopes, referred to as Rendzic Leptosols and Hypereutric Epileptic Cambisols, are more superficial and have higher rock fragment contents. Stagnosols and Gleysols can be found at the bottom of the valleys and over the Kimmeridgian marls and limestones: they are deep, clay-rich and hydromorphic soils; the former is waterlogged for part of the year while the latter is waterlogged all year round. In the north-east of the study area, clay-rich and $CaCO_3$-bearing materials from a tunnel excavation in 1841-1846 form local pockets of Technosols, which were not considered in this study due to their limited spatial extent. Land use information was derived from the 1/100,000 CORINE Land Cover 2018 at a resolution of 25 ha.

The data from the 86 soil profiles contain SOC content data in the different soil horizons (253 data points), but only 48 bulk density measurements using the cylinder method. In order to provide additional SOC content and bulk density data as a function of depth, 112 additional profiles corresponding to these eight soil types were collected from soil databases in the six surrounding administrative geographical units (counties). The soil profiles were collected by the RMQS (French Soil Quality Monitoring Network) and Renecofor (French Permanent Plot Network for the Monitoring of Forest Ecosystems). In each collected sample, organic carbon content (g kg$^{-1}$) is measured in the fine fraction (< 2 mm) by dry combustion after removal of the inorganic carbon with acid. Since this study only considers mineral soil, the litter layer was excluded from the forest profiles. Bulk density values are measured using the cylinder method in 552 out of the 932 samples, and are otherwise estimated from a pedotransfer function from Beutler *et al.* (2017) based on clay and total organic content values as follows:

$$BD = [1.6179 - 0.0180 * (Clay + 1)^{0.46} - 0.0398 * SOC^{0.55}]^{-1.33} \qquad (1)$$

where BD is the bulk density (kg m$^{-3}$), Clay is the clay content (g kg$^{-1}$), and SOC is the total organic carbon content (g kg$^{-1}$).





**Figure 1: Land uses, soil types and geomorphological context of the study region. (a) Land use (Source: Corine Land Cover 2018). (b) Map of dominant soil types (Source: Party *et al.*, 2019). (c) Synthetic cross-section of the geology, topography and dominant soil types in the region of study.**




**Table 1: Mean values of pH, clay content, rock fragments content and CaCO₃ concentration for each soil type and horizon, calculated from 86 whole soil profiles sampled between 1995 and 2019 within the region of study. Standard deviations are given in brackets.**

| Soil Type | Horizon | Depth (cm) | Horizon Thickness (cm) | Clay (g kg⁻¹) | pH | Rock fragments (%) | CaCO₃ (g kg⁻¹) |
|---|---|---|---|---|---|---|---|
| Calcaric Rendzic Leptosols | 1 | 35 (9) | 16 (5) | 478 (68) | 7.8 (0.9) | 3 (15) | 58 (118) |
| | 2 | | 19 (6) | 392 (123) | 8.3 (0.4) | 35 (30) | 414 (186) |
| Calcaric Cambisol | 1 | 60 (17) | 14 (6) | 462 (110) | 7.8 (0.9) | 8 (15) | 13 (136) |
| | 2 | | 21 (11) | 394 (87) | 8.2 (0.4) | 35 (23) | 465 (250) |
| | 3 | | 25 (11) | 328 (110) | 8.3 (0.3) | 70 (20) | 389 (246) |
| Hypereutric epileptic Cambisol | 1 | 43 (11) | 22 (7) | 489 (73) | 7.8 (0.8) | 0 | 0 |
| | 2 | | 21 (5) | 523 (86) | 6.9 (1.1) | 60 (31) | 0 |
| Hypereutric Cambisol | 1 | 84 (61) | 20 (6) | 409 (125) | 6.9 (1.0) | 2 (13) | 0 |
| | 2 | | 30 (14) | 522 (147) | 7.5 (0.7) | 3 (28) | 0 |
| | 3 | | 33 (45) | 733 (119) | 7.8 (0.4) | 50 (26) | 2 (5) |
| Eutric Cambisol | 1 | 85 (30) | 18 (6) | 278 (107) | 5.6 (0.8) | 0 | 0 |
| | 2 | | 27 (10) | 484 (164) | 6.2 (1.0) | 0 | 0 |
| | 3 | | 40 (28) | 595 (207) | 7.5 (1.5) | 5 (36) | 2 (17) |
| Dystric Cambisol | 1 | 168 (33) | 15 (5) | 40 (1) | 4.0 (0.2) | 0 | 0 |
| | 2 | | 18 (3) | 27 (6) | 4.3 (0.2) | 0 | 0 |
| | 3 | | 10 (0) | 40 (8) | 4.3 (0.2) | 0 | 0 |
| | 4 | | 48 (3) | 75 (9) | 4.7 (0.1) | 0 | 0 |
| | 5 | | 78 (23) | 95 (44) | 4.6 (0.1) | 0 | 0 |
| Stagnosol | 1 | 115 (30) | 28 (5) | 490 (182) | 7.8 (1.0) | 0 | 2 (196) |
| | 2 | | 40 (11) | 353 (131) | 8.2 (1.4) | 0 | 98 (244) |
| | 3 | | 47 (11) | 346 (111) | 8.4 (1.2) | 1 (15) | 576 (236) |
| Gleysol | 1 | 140 (41) | 23 (7) | 453 (88) | 7.8 (0.4) | 0 | 103 (105) |
| | 2 | | 46 (12) | 386 (62) | 8.2 (0.3) | 0 | 143 (189) |
| | 3 | | 72 (36) | 350 (75) | 8.2 (0.3) | 0 | 290 (288) |

## 2.2 Estimation of current and maximum SOC stocks

### 2.2.1 Current SOC stocks

Soil organic carbon stocks per surface unit are calculated as follows (Chen *et al.*, 2019a):

$$\text{SOCstock} = \frac{p * \text{SOC} * \text{BD} * (100 - \% \text{ Rock fragments})}{1000} \tag{2}$$

where SOC$_{stock}$ is the total SOC stock (kg m⁻²), p is the actual thickness (m) of topsoil or subsoil, SOC the soil organic carbon content (g kg⁻¹), BD the bulk density (kg m⁻³ = g dm⁻³) and % Rock fragments the percentage of elements > 2 mm (%).

This methodology assumes that the fraction > 2 mm does not contain organic carbon, which has been disputed by Harrison *et al.* (2011) in cases where the rock fragments are abundant and display signs of porosity and weathering.



The median soil organic carbon content (SOC in g kg$^{-1}$) as a function of depth for each soil type and land use was calculated using the typical SOC content profile established by Mathieu *et al.* (2015) and Jreich (2018) on the basis of three descriptors:

$\Omega_1$ the SOC content of the soil type at maximal depth, $\Omega_2$ the SOC content at the surface, and $\Omega_3$ the depth at half maximum of the SOC content:

$$\text{SOC}\,(s,z) \;=\; \Omega1(s) \;+\; (\Omega2(s) - \Omega1(s)) \, * \, e^{-(z/\Omega3(s))} \tag{3}$$

where *s* is the soil type, *z* the depth.

This method was used to interpolate SOC content data from national and regional datasets, acquired per horizon, in order to
obtain the continuous distribution of SOC stock over the whole soil profile for each soil type and land use considered. A least square method for non-linear curve-fitting (Matlab function lsqcurvefit) was then applied to adjust the $\Omega_{1-3}$ parameters (Appendix Table 2).

Continuous vertical profiles of median bulk density were then obtained for each soil type using a logarithmic fit. The horizon thickness and percentage of rock fragments correspond to the median of the values per horizon per soil types in the 86 profiles
within the OPE zone. The median SOC stock is then calculated at each 1 cm interval along the whole profile based on the median bulk density curve, the median SOC curve and the percentage of rock fragments.

### 2.2.2 Maximum SOC stocks

We call maximum SOC stocks the highest SOC stocks that a given soil type is estimated to be able to contain. The maximum SOC additional storage capacity corresponds to the difference between the maximum SOC stocks and the current SOC stocks
(corresponding to the year 2018). Two methods were compared to estimate the maximum SOC stocks: the carbon saturation method using the fine fraction of the soil (clay and fine silt) (Hassink, 1997), and a data-driven approach using the upper percentiles of measured SOC content data (Chen *et al.*, 2019a). We call the first method the Hassink equation and the second method the data-driven approach.

The Hassink equation was established empirically on the basis of 20 Dutch grassland soils considered to be at the stationary
state. Further studies then upgraded the model by considering a greater range of land uses and clay types (Six *et al.*, 2002), and by selecting the upper percentiles of the dataset to avoid the inclusion of undersaturated soils (Feng *et al.*, 2013; Georgiou *et al.*, 2022). Fernández-Catinot *et al.* (2023) compared those different equations in the topsoil and found the lowest estimates of SOC saturation in MAOM with the equations by Six *et al.* (2002) and Hassink (1997). The Hassink equation is as follows:

$$\text{Csat} \;=\; 4.09 \;+\; 0.37 \, * \, (\text{Clay} + \text{fineSilt})\ (\%) \tag{4}$$

where $C_{sat}$ is the theoretical carbon saturation concentration in the fine fraction in g kg$^{-1}$.

The SOC saturation deficit corresponds to the space still available in the fine fraction for binding carbon, calculated as the difference between $C_{sat}$ and the current SOC content in the fine fraction ($C_{fine}$). Since our dataset does not distinguish between POM and MAOM, $C_{fine}$ is estimated at 85 %, 66 % and 69 % of the total SOC content in cropland, forest and grassland topsoils, and 93 %, 75 % and 86 % for cropland, forest and grassland subsoils respectively, with the limit between topsoils and subsoils
set at 30 cm, following Angers *et al.* (2011) and Chen *et al.* (2018).





The data-driven approach estimated the maximum SOC stocks using a depth-dependent boundary line based on the 75[th] percentile of the measured SOC stocks data points. Since the number of SOC data point per soil type ranges from 29 (Hypereutric Epileptic Cambisol) to 268 (Stagnosol), the logarithmic regressions used to estimate the maximum SOC stocks are based on a minimum of 7 and up to 67 data points. That boundary line was considered to represent the maximum SOC

stock that a given soil type can reach under the land management strategies represented in the region of study. The maximum SOC stock is therefore region-dependent as it is not solely driven by the intrinsic textural properties of the soil, but also by climate and plant productivity as they influence soil biology and chemistry along the soil profile.

The choice in percentile value strongly affects the estimation of maximum SOC stocks (Chen *et al.*, 2019a). To explore this impact and feed a discussion on criteria for percentile selection, we calculated SOC maximum stocks at the 88[th] percentile by

selecting the data points above the 75[th] percentile boundary line and fitting another logarithmic regression to these upper values.

A bootstrap method was used to determine the overall uncertainty of the current SOC stocks and maximum additional storage capacity for each soil type at 90% confidence interval (Chen *et al.* 2019a). In this method, we generate random subsets of input parameters SOC, BD, percentage of rock fragments and depth values within the standard deviation of each soil type, and repeat

the procedure 1000 times to obtain 1000 estimates of the mean and percentiles values of the carbon stocks.

## 2.3 Simulation of SOC accrual at different timescales

We applied a whole-profile SOC dynamic model to simulate the vertical repartition of SOC accrual over different timescales. The approach is illustrated in Figure 2, with further details of model functioning and equations in Appendix 1.

The profiles of current SOC stocks were first discretized into 10 cm layers. For each layer, the current SOC stock was allocated

to three pools (fast, slow, stable) corresponding to different SOC mineralization rates defined by Balesdent *et al.* (2018) based on a meta-analysis of changes in stable carbon isotope signatures at 55 tropical grassland, forest and cropland sites. The mineralization rates were obtained using a $C_3/C_4$ approach, which is typically very efficient to follow carbon dynamics over timescales ranging from one to one thousand years. This timescale enables exploring the impacts of land use change on SOC dynamics, which make the $C_3/C_4$ approach more relevant for land planning compared to the $^{14}C$ method, which covers

timescales of several thousand years (Verma *et al.*, 2017).

The SOC mineralization factors associated with each pool were then corrected for temperate soils using correction factors defined for the AMG model to account for the differences in environmental conditions (temperature and humidity) between tropical and temperate, but also to account for the differences in pH, clay content and $CaCO_3$ between soil types (Mary *et al.*, 1999; Clivot *et al.* 2017; Levavasseur *et al.*, 2020). The mean residence times as a function of depth derived from the corrected

mineralization factors in the fast and slow pools can be found in Appendix Table 3.

The whole-profile SOC dynamic model was initialized under the assumption that the current SOC stocks in 2018 were at steady state. This assumption was justified on average by a land occupation map from 1830 showing limited changes in land use over the past 200 years (Dupouey *et al.*, 2008). Inversing the model at the steady state yielded the vertical repartition of



yearly C inputs (See Appendix Table 4), which we call stationary C inputs in Figure 2. The stationary C inputs were in
agreements with estimations derived from the method of Bolinder *et al.* (2007) based on crop yield (Appendix Figure 1).

The maximum SOC stocks estimated by the 75[th] percentile data-driven approach were then used as the target for each soil type. We call maximum C inputs the vertical repartition of annual inputs needed to reach and maintain this target stock (Figure 2). The difference between the maximum and stationary C inputs corresponds to the additional C inputs necessary to reach the maximum SOC stocks. The additional C inputs vary with soil types and especially with land use, with mean additional C inputs
of 0.5 tC ha$^{-1}$ y$^{-1}$ under forests, 1.0 tC ha$^{-1}$ y$^{-1}$ under grasslands and 1.5 tC ha$^{-1}$ y$^{-1}$ under croplands (Figure 2). These values correspond to a theoretical scenario wherein soils under all land uses reach the maximum SOC stock specific to their soil type. This scenario will be confronted in the discussion section against values of C input and SOC accrual found in the literature following a change in land management practice, from conventional to conservation agriculture for instance. By contrast, reaching the maximum SOC stocks estimated by the 88[th] percentile required additional C inputs of about 3.4 tC ha$^{-1}$ y$^{-1}$ across
all land uses. This was not considered realistic, as will be justified in the discussion section.

The equations of SOC stock evolution over time were then applied for this scenario of additional C inputs dependent on land use. The scenario was run over 5000 years to visualize the new steady state, though we were mostly interested in the SOC accrual reached after 25 years to fit with the carbon neutrality timeline.

Finally, we tested the effect of projected rises in temperature on the simulated SOC accrual by modifying the mineralization
correction factor linked to temperature in the AMG model (see Appendix Equation 1). The temperature was increased linearly to projected annual temperatures in metropolitan France in 2050 according to the scenarios RCP4.5 and RCP8.5 (+1.0°C and +1.3 °C from the mean temperatures of 1991-2020 respectively based on Soubeyroux *et al.*, 2020). This corresponds to an increase in mean annual temperatures from 10.4 °C to 11.4°C (RCP4.5) or 11.9°C (RCP8.5) over 25 years at all depths. The 1.0 °C increase in temperature in the region of study under scenario RCP4.5 was corroborated by model simulations of mean
annual temperatures by the Meteo France ALADIN63_CNRM-CM5 model within an 8 km radius area around Bure (55087), comparing the year intervals 2046-2055 and 2009-2019 (Drias, données Météo-France, CERFACS, IPSL). RCP8.5 amounts to an extreme scenario in terms of increased mineralization rates, since in addition to using the most pessimistic RCP scenario, our model assumes that rises in temperature propagate instantly at depth and that humidity conditions remain at the present levels. We tested the sensitivity of SOC accrual to the two temperature scenarios in the different soil types and land covers.





**Figure 2: Illustration of the approach used to simulate SOC accrual over different timescales. (a) Estimation of the vertical repartition of current and maximum SOC stocks. (b) Estimation of maximum and stationary C inputs at each depth by model inversion of the maximum and current SOC stocks. (c) Estimation of additional C input scenarios averaged per land use for all soil types. (d) Application of the scenarios of C inputs at each depth. The stationary C inputs and maximum C inputs are provided in Appendix Table 4.**







## 2.4 Spatialization of SOC stocks and maximum SOC additional storage capacity

The study site was divided into zones characterized by their land use (cropland, grassland, forest) and by their dominant soil type. Zones were derived from the intersection of the CORINE Land Cover map and of the pedological map. The soil units represented in Figure 1b show the dominant soil type in each cartographic zone, but in reality, each zone contains a non-spatialized mixture of soil types. Therefore, the SOC stock and SOC additional storage capacity in each cartographic zone correspond to a weighted mean following the percentage of each soil type present in the zone. Likewise, the standard deviation of SOC stocks in a zone corresponds to a weighted mean of the standard deviation of the SOC stock in each represented soil type.

Additionally, the non-spatialized repartition of soil types causes a local uncertainty since SOC stock can be expected to change abruptly at the boundary between two soil types. We visualize this uncertainty by mapping the weighted differences between the mean stock of each represented soil type and the mean stock within the zone (Appendix Figure 2).

## 3 Results

## 3.1 SOC stock and maximum additional storage capacity a function of depth, land use and soil type

### 3.1.1 Vertical repartition of SOC stocks

Current SOC stocks over the whole profile range from 78 to 333 tC ha$^{-1}$ (Table 3), of which 59 to 156 tC ha$^{-1}$ are in the topsoil (0 - 30 cm). The lowest SOC stocks are found in the shallower soil types (Calcaric Rendzic Leptosol and Hypereutric Epileptic Cambisol). Current SOC stocks are twice to three times higher in hydromorphic soils (Stagnosols and Gleysols) compared to non-hydromorphic soils.

SOC content and stocks decrease with depth, with sharp decreases in the SOC stock profiles corresponding to a change in the percentage of rock fragments between two horizons (Figure 3a-c). On average, excluding the shallower soil types (Calcaric Rendzic Leptosol and Hypereutric Epileptic Cambisol), the proportion of the SOC stock situated in the first 30 cm is 53 % in croplands, 67 % in grasslands and 71 % in forests. The soils in croplands are therefore depleted in SOC in the topsoil compared to forests and grasslands (Figure 3a). The difference in SOC stocks between land uses diminishes in the deeper horizons.





**Table 2: Soil organic carbon stocks, maximum stocks corresponding to the 75th percentile of the datasets under all land uses, and maximum SOC additional storage capacity estimated from the data-driven approach for the different land uses and soil types represented, over the whole profile (for the stocks and additional storage capacity above and below 30cm, see Appendix Table 5). The 90% confidence intervals of current stocks as determined by the bootstrap method are provided in brackets.**

| | Median SOC stocks in 2018 (tC ha$^{-1}$) | | | Maximum stocks (75th percentile) (tC ha$^{-1}$) | Maximum SOC additional storage capacity (tC ha$^{-1}$) | | |
|---|---|---|---|---|---|---|---|
| | Cropland | Grassland | Forest | All land uses | Cropland | Grassland | Forest |
| **Calcaric Rendzic Leptosol** | 78 (48 - 115) | 101 (84 - 138) | 149 (97 - 183) | 167 | 89 | 66 | 19 |
| **Calcaric Cambisol** | 100 (58 - 133) | 134 (66 - 183) | 148 (104 - 184) | 191 | 91 | 57 | 44 |
| **Hypereutric Epileptic Cambisol** | 92 (49 - 129) | | 106 (76 - 121) | 129 | 37 | | 22 |
| **Hypereutric Cambisol** | 103 (62 - 137) | 167 (125 - 255) | 160 (92 - 204) | 228 | 125 | 60 | 68 |
| **Eutric Cambisol** | 102 (59 – 144) | 90 (66 - 115) | 157 (71 - 190) | 194 | 93 | 105 | 38 |
| **Dystric Cambisol** | | | 120 (76 - 198) | 169 | | | 49 |
| **Stagnosol** | 166 (101 - 237) | 161 (108 - 279) | 172 (121 - 249) | 285 | 119 | 124 | 113 |
| **Gleysol** | 279 (154 - 417) | 333 (252 - 466) | | 476 | 197 | 143 | |



**Figure 3: (a) Median (50th percentile of the dataset) fitted depth profiles of SOC content in each soil type and each land use. The Jreich parameters (2018) used to plot the SOC content profiles are given in Appendix Table 2. (b) Estimation of maximum SOC content as a function of depth by the Hassink equation corrected with proportions of POM and MAOM from the literature (dashed blue line) and by the data-driven approach using a 75th percentile curve of the dataset (black solid line). (c) Current SOC stocks under croplands and maximum SOC additional storage capacity to reach the maximum SOC stocks of each soil type.**





### 3.1.2 Maximum SOC stocks and additional storage capacity

The maximum SOC content estimated by the data-driven approach decrease with depth under all soil types, from 50-100 g kg$^{-1}$ near the surface to under 25 g kg$^{-1}$ at the bottom of the soil profiles (Figure 3b-c). By contrast, the decrease in maximum SOC content with depth as estimated by the Hassink equation is less prominent. Maximum total SOC contents stay at around 35 - 46 g kg$^{-1}$ throughout all soil profiles (32 - 40 g kg$^{-1}$ under 30 cm), except for the Dystric Cambisol where the average value is 11 g kg$^{-1}$ (Figure 3b). The maximum SOC stocks as estimated by the data-driven approach range from 129 tC ha$^{-1}$ in the Hypereutric Epileptic Cambisol to 476 tC ha$^{-1}$ in the Gleysols.

The maximum SOC additional storage capacity found by the data-driven approach varies from 19 tC ha$^{-1}$ for shallow, rocky forest soils to 197 tC ha$^{-1}$ for agricultural Gleysols (Table 2), considering the conversion of cropland into grassland or forest. Using percentile 88th instead of 75th increases our estimation of the maximum SOC stocks by about 16% (9 - 27% depending on soil type), without changing the hierarchy of maximum SOC stocks across the eight soil types.

According to the 75th percentile method, soils in the region of study are at 74% of their maximum SOC stock on average, ranging between 16-61% for croplands, 30-56% grasslands and 40-82% for forests. Across all land uses, the shallow rocky soils (Calcaric Rendzic Leptosol and Hypereutric Epileptic Cambisol) are closer to their maximum SOC stocks than the Stagnosols and Gleysols.

### 3.2 Exploring kinetics of simulated SOC accrual

Employing the Hassink equation was considered to underestimate the maximum SOC content near the surface and to overestimate maximum SOC content in the deeper horizons, for reasons that will be detailed further in the Discussion section. The maximum SOC content profiles obtained by the data-driven approach were therefore selected for the exploration of SOC kinetics.

The equations of our model calculate the SOC mean residence times per depth as a function of the physico-chemical properties of the studied soil types (see Appendix equations 1-5). In our study site, they range from 50 – 100 years above 30 cm and from 145 – 453 years below 30 cm (Appendix Table 3). The increase in mean residence time with depth is stark in the slow pool (477-1100 years in the first 10 cm to 863-5817 years in the deeper soil horizons), but is not visible in the fast pool (17-38 years in the first 10 cm to 11-47 years in the deeper soil horizons) (Appendix Table 3). Since most of the new C inputs is allocated to the fast carbon pool and in the surface horizons (Appendix Table 3-4), the SOC accrual is not strongly affected by soil type over 25 years.

The stationary C inputs obtained by model inversion ranged from 1.0 – 4.6 tC ha$^{-1}$ y$^{-1}$, and the maximum C inputs from 1.4 – 6.0 tC ha$^{-1}$ y$^{-1}$ (Appendix Table 4). Under the chosen scenario of C inputs dependent on land use (+1.5 tC ha$^{-1}$ y$^{-1}$ under cropland, +1.0 tC ha$^{-1}$ y$^{-1}$ under grassland, +0.5 tC ha$^{-1}$ y$^{-1}$ under forest), and when rising temperatures are not considered, the SOC accrual after 25 years ranges from 22-26 tC ha$^{-1}$ under cropland, 15-18 tC ha$^{-1}$ under grassland, to 8-10 tC ha$^{-1}$ under forest (Figure 4, Appendix Table 6). The yearly accrual rate averaged over the first few decades is therefore 0.88-1.04 tC ha$^{-1}$





$y^{-1}$ under croplands, 0.6-0.72 tC ha$^{-1}$ y$^{-1}$ under grassland and 0.32-0.4 tC ha$^{-1}$ y$^{-1}$ under forest. The accrual rates then decrease over decadal and centennial timescales as the SOC stocks stabilise asymptotically towards the new steady state, as per the model equations. SOC accrual at the new steady state is highest under Dystric Cambisol owing to the effect of the low pH on

325 the mineralization rates as implemented in the model. Modelled SOC accrual after 25 years decreases with depth under all soil types and land uses (Figure 5).

Under the RCP4.5 scenario of 1.0 °C increase over 25 years, the SOC accrual is attenuated by 7 to 38% compared to the accrual simulated at constant temperature (10% under cropland, 20% under grassland and 30% under forest on average). The SOC accrual after 25 years under this scenario ranges from 16-24 tC ha$^{-1}$ under cropland, 10-16 tC ha$^{-1}$ under grassland, to 5-

330 8 tC ha$^{-1}$ under forest (Appendix Table 6).

Incorporating the more extreme RCP8.5 scenario of 1.3 °C increase in temperature over 25 years attenuates SOC accrual by 10 to 50%, and shows a stronger impact of soil type and especially land cover on the mineralization rates (Appendix Table 6). SOC accrual is attenuated by 10-20% in cropland soils, 10-40% in grassland soils, and 40-50% in most forest soils except Dystric Cambisols (20%).

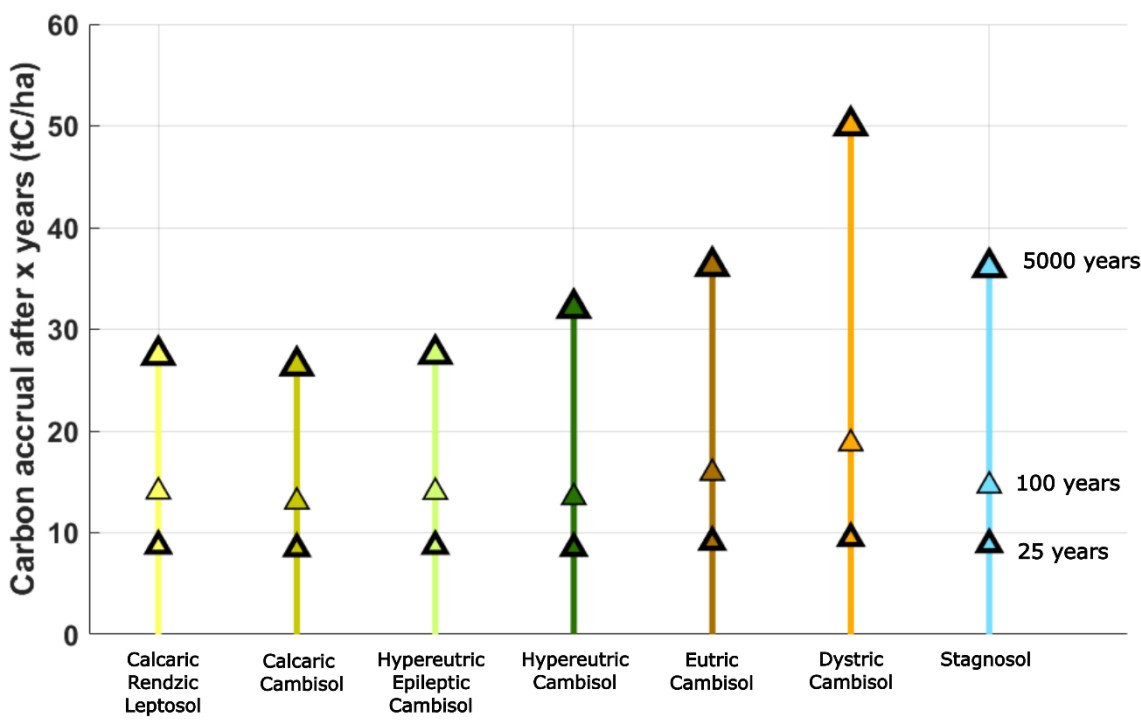

335

**Figure 4: Model results of SOC accrual after 25, 100 and 5000 years under forests for a scenario of +0.5 tC ha$^{-1}$ y$^{-1}$ compared to the stationary C inputs.**



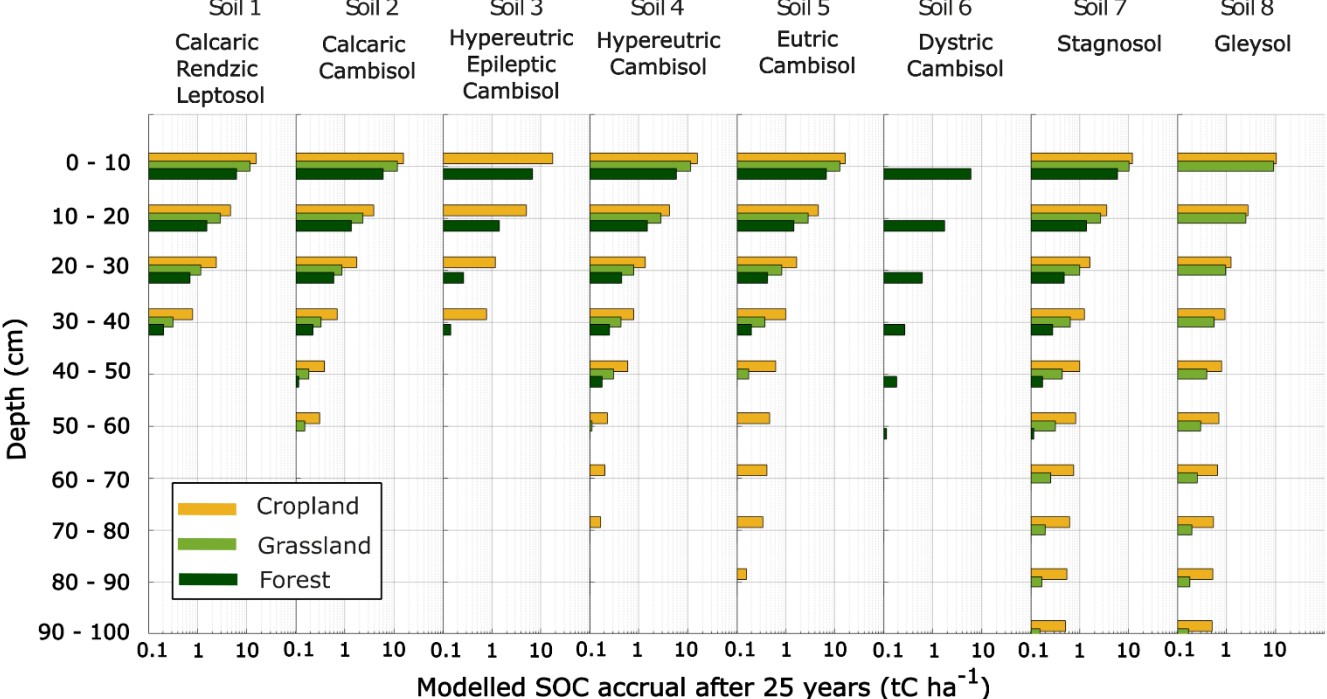

**Figure 5: Model results of SOC accrual after 25 years at each depth under the three considered C input scenarios (+1.5 tC ha$^{-1}$ y$^{-1}$ under croplands, +1.0 tC ha$^{-1}$ y$^{-1}$ under grasslands, +0.5 tC ha$^{-1}$ y$^{-1}$ under croplands compared to the stationary C inputs)**

### 3.3 Maps of SOC stocks, maximum additional storage capacities, and simulated accrual after 25 years

The repartition of SOC stocks and maximum additional storage capacities in the region of study are most visibly related to the land use, but is also affected by the spatial distribution of Stagnosols and Gleysols (Figure 6). The current SOC stock in the region of study amounts to a total of 3.9 MtC, with a standard deviation of 1.5 MtC according to the bootstrap method (Appendix Figure 3). To compare these results with national-scale estimates of SOC stocks, we average 3.9 MtC over the entire region of study and obtain a mean value of 122 tC ha$^{-1}$, of which 87 tC ha$^{-1}$ are in the first 30 cm.

The maximum SOC stocks that the region can theoretically contain is 3.9 + 2.5 = 6.4 MtC, suggesting that the soils in the region of study are at 61% of their theoretical maximum SOC stock. However, according to model results in the chosen scenario, this maximum SOC stock would only be reached over timescales of centuries to millenia, and the SOC accrual after 25 years only reaches 0.57 MtC. The SOC accrual in the region of study is attenuated by 14% and reaches 0.49 MtC when a 1.0 °C increase in temperature is implemented in the mineralization rates (Appendix Figure 4).



**Figure 6: Maps of SOC stocks (a), maximum SOC additional storage capacity (b) and SOC accrual after 25 years under a scenario of additional C inputs compared to the steady state dependent on land use (+ 0.5 tC ha⁻¹ y⁻¹ under forests, + 1.0 tC ha⁻¹ y⁻¹ under grassland, + 1.5 tC ha⁻¹ y⁻¹ under cropland) (c). Upper and lower confidence intervals provided by the bootstrap method are given in Appendix Figure 3. Standard deviation of the total SOC stocks and additional storage potentials based on the upper and lower confidence intervals applied to the whole region is 1.5 MtC.**





## 4 Discussion

**4.1 Maximum additional storage capacity of total soil organic carbon across depths**

Several approaches have been proposed to quantify the maximum stocks of soil organic carbon (SOC) that can be contained or stabilized over the whole soil profile: modelling for a selected management option, the carbon saturation in the fine fraction, typically estimated by the Hassink equation, and the maximum SOC stock for a given management strategy, estimated by a data-driven approach (Barré *et al.*, 2017; Chen *et al.*, 2019a). The data-driven approach does not consider that SOC stocks

have a finite limit but assumes an equilibrium dependent on C inputs, biological activity and soil physico-chemical properties. The processes by which organic matter binds itself to mineral surfaces, and whether the mineral surface can saturate in organic matter, is still an active subject of scientific investigation (Schweizer, 2022, Begill *et al.*, 2023, Fernández-Catinot *et al.*, 2023, Six *et al.*, 2024). The pedon-scale of our research does not provide further understanding of SOC sequestration mechanisms, but rather explores the adequacy of applying these two methods to the whole soil profile.

Exploring total SOC content distributions with a fine vertical resolution highlights their rapid diminution with depth. SOC contents are divided by two to four between the top and bottom of the profile (50-100 g kg$^{-1}$ in the first 10 cm versus 25 g kg$^{-1}$ at depth), with the steepest decrease occurring within the first 30 cm under grassland and forest. Maximum SOC contents estimated by the data-driven approach show similar vertical trends. By contrast, the continuous profiles of SOC saturation based on the Hassink equation show only a limited decrease with depth, staying around 32 - 40 g kg$^{-1}$ under 30 cm for most

soil types. This is because the Hassink equation only provides a theoretical maximum based on a textural limit, without accounting for variations in C inputs with depth.

Previous studies found a theoretical maximum SOC content in the MAOM fraction of 40-50 g per kg of soil, and considered this theoretical maximum to be applicable at all depths (Cotrufo *et al.*, 2019, Georgiou *et al.*, 2022). However, in our region of study, the maximum SOC content found by the data-driven approach at the bottom of the soil profiles did not exceed 25 g

per kg of soil (Figure 3a), including both POM and MAOM and including forest soils. This is similar to the discrepancy between Hassink and data-driven results in subsoils observed by Chen *et al.* (2019b). We therefore consider that increasing SOC content above this value, while theoretically possible from the perspective of the clay and silt content, is unrealistic at depth owing to the biotic and abiotic controls of carbon dynamics. The spatially resolved data-driven approach yields vertical repartitions of maximum SOC stocks that are more in line with what we know of root distribution and therefore C input

distribution (Joggáby & Jackson 2000). The data-driven approach implicitly accounts for the biological functioning of the different soil types: the maximum SOC stocks derived from this method result from an equilibrium between the primary C inputs and the decomposition of SOC, which depends on the physico-chemical properties of each soil type.

A difficulty of using the data-driven approach to estimate maximum SOC stocks is that the percentile regression necessarily depends on the size of the dataset and on its variability. A low percentile value within a large dataset underestimates the storage

capacity, but an overly high percentile value within a small dataset produces an unrealistic target. Indeed, certain soils profiles benefit from specific management (ie. fenced forested plots preventing grazing) or physico-chemical conditions (ie. deep



anoxic, carbon-rich horizons) that locally increase their SOC stock. An overly high percentile value yields as target an artificial soil profile whose high SOC stock results from an unachievable combination of pedological and management characteristics. In this regional study, using the 75th percentile of the dataset (198 profiles) led to C input scenarios of 0.5 tC ha$^{-1}$ y$^{-1}$ under

forests, 1.0 tC ha$^{-1}$ y$^{-1}$ under grasslands and 1.5 tC ha$^{-1}$ y$^{-1}$ under croplands. Using the 88th percentile led to C input scenarios of 3.4 tC ha$^{-1}$ y$^{-1}$ across all land uses. Our use of a vertically resolved regression did not allow us to test for intermediate percentile values. However, the 75$^{th}$ percentile did provide C input scenarios compatible with values found in the literature in cropland, forest and grassland soils following land management change, as will be detailed in section 4.3, and so was considered to be an acceptable target.

The data-driven approach is currently applied with varying percentile values and at various spatial scales: Chen *et al.* (2019a) compared maximum total SOC stocks following the 0.8, 0.85 and 0.9 percentile value at the national scale (1089 sites). Georgiou *et al.* (2022) compared the maximum mineral-associated SOC with low and high activity minerals at the 0.9, 0.95 and 0.975 percentiles at the global scale (1144 profiles). Standardized rules to define the choice of a percentile value depending on the scale of the study and the size and variability of the dataset have yet to be established.


### 4.2 Carbon accrual across depths over decadal timescales

Modelled SOC accrual ranged from 8.5 to 26 tC ha$^{-1}$ after 25 years, with a rapid decrease of SOC accrual rates with depth driven by decreasing C inputs. This implies that the deeper horizons of the soil provide limited opportunity for additional storage over short timescales using current land management practices. Furthermore, the proportion of new carbon inputs that

is allocated to the fast carbon pool exceeds 85% at all depths in the soil profile (Appendix Table 3), meaning that, even in the deeper soil horizons, the majority of new C inputs is quickly mineralized, as also simulated by Sierra *et al.* (2024). The fast pool in the deeper soil horizons has estimated mean residence times of 11 - 47 years, while the slow pool has mean residence times of 1744 - 5817 years. Contrastingly, in the first 10 cm, SOC mean residence times in the fast pool are 17 - 38 years while the slow pool has mean residence times of 477 - 1100 years. The greater contrast in mean residence times between the fast and

slow pools at depth challenges our understanding of SOC dynamics.

While our model provides a widely-applicable tool to assess the effect of different soil types and initial distributions of SOC stocks on SOC dynamics at decadal timescales, it remains to be validated with measures of SOC accrual using repeated sampling campaigns, and does not cover all processes relevant to organic matter dynamics in soils. The equations of the model imply that the SOC stock is linear with the C inputs, and the effect of soil physico-chemical properties on mineralization rates

is approximated by correction factors for pH, clay content and CaCO$_3$ (Appendix Equations 3-5). The priming effect, however, is not taken into consideration, which could lead the simulated results to overestimate SOC accrual (Guenet *et al.*, 2018).

Testing for the effect a +1.0°C increase in temperatures on mineralization rates led to an attenuation of SOC accrual by 2050 of 14 % over the region of study (Appendix Figure 4), but our model could not account holistically for the effects of climate change on SOC dynamics. Those effects are expected to be very context-dependent since the combination of changes in



temperature, $CO_2$ concentration and precipitation can drive a myriad of responses in net primary production, SOC input repartition and mineralization processes (Rocci *et al.*, 2021; Bruni *et al.*, 2021). In forests for instance, increased drought conditions may increase tree mortality, but might also enhance deeper roots prospection for water, thereby changing the vertical repartition of C inputs (Schlesinger *et al.*, 2016). Here, we have considered that humidity conditions would not change from the 2009-2019 period and would not affect soil carbon dynamics in the region of study (see Appendix Equation 2). Modelling

of future temperatures and precipitations using the Drias model under scenario RCP4.5 indicate a weak increase in annual accumulated rainfall (+2%), which considering the +1.0°C temperature increase could lead to lower soil humidity.

Furthermore, recent studies suggest that the vulnerability of SOC to rising temperatures depends on soil textural properties, with fine-textured soils being less sensitive to climate warming (Hartley *et al.*, 2021). The scientific community needs to improve its understanding of the priming effect, of SOC dynamics processes driven by climate change, and to further explore

how soil type influences organic matter decomposition dynamics over decadal timescales.

Soil type did not appear to play an important role on SOC accrual over short timescales (Figure 4): the differences in mineralization rates across soil types are not sufficient to have a significant impact after 25 years, especially in the fast pool (Appendix Table 3). It is rather the land use that affects SOC accrual by controlling the quantity and vertical repartition of inputs (Appendix Table 4). However, soil type has a strong influence on current SOC stocks by categorizing soils based on

soil depth, rock fragment content and other physico-chemical properties. Hydromorphic soils in particular have total SOC stocks up to three times that of other soil types, making their preservation particularly critical. These high SOC stocks are due to waterlogged conditions strongly limiting decomposers activity (Sahrawat, 2004), notably for energetic reasons (Keiluweit *et al.*, 2016). The current SOC stocks have been built over timescales of centuries to millennia, especially in the deeper soil horizons, but can be rapidly lost due to land use change and other disturbances. Therefore, as highlighted by Sierra *et al.*

(2024), the priority should be to preserve the existing SOC stocks, even as we attempt to implement innovative land management practices to maximise these SOC stocks.

**4.3 Opportunities and challenges for soil contributions to carbon neutrality**

An important aspect of this work is how much additional carbon can be added to soils over the decadal timescales relevant to

stakeholders. The maximum soil organic carbon (SOC) additional storage capacity can be used as a theoretical, long-term target value, but is not representative of how much carbon can realistically be added to soils over decadal timescales. In the region of study, total SOC accrual after 25 years was found to be five times lower than the SOC additional storage capacity (0.57 MgC versus 2.5 MgC). Nevertheless, we observe potential for SOC accrual over 25 years in all our studied soils. Our simulation of rising temperatures following RCP4.5 (+1.0 °C) and RCP8.5 (+1.3°C) attenuated this SOC accrual by 7 - 38%

and 10 - 50% respectively over 25 years through the increase of mineralization rates. This shows that increasing organic matter inputs to the soil remains worthwhile, since SOC accrual remains significant even in an extreme scenario (most pessimistic conditions of temperature increase plus optimal humidity conditions).





The scenarios of C inputs and resulting modelled rates of SOC accrual are broadly in line with monitoring results found over decadal timescales following land management change. The 1.5 tC ha$^{-1}$ y$^{-1}$ additional C inputs in cropland are close to values

calculated in a long-term field experiment after transition from conventional agriculture to conservation agriculture (1.72 tC ha$^{-1}$ y$^{-1}$ over 16 years, Autret *et al.*, 2016). Under grasslands, a 1.0 tC ha$^{-1}$ y$^{-1}$ increase in C inputs is reachable by increasing plant diversity: the resulting SOC accrual rates of 0.6 – 0.7 tC ha$^{-1}$ y$^{-1}$ over the entire soil profile in the first 25 years (Appendix Table 6) are in line with measured increases in C accrual rates of 0.34 tC ha$^{-1}$ y$^{-1}$ in the first 20 cm over 22 years following an increase in grassland plant diversity (Yang *et al.*, 2019). Under forests, strategies to increase SOC stocks include less frequent

cutting, acting on forest productivity to increase root inputs and limiting soil disturbance during harvesting (Jandl *et al.*, 2007; Mayer *et al.*, 2020). A study of SOC accumulation in the first 15 cm of the mineral soil under rotation lengths in humid temperate forests ranging from 10 to 20 years found an increase in SOC accrual rates of 0.42 to 0.61 tC ha$^{-1}$ y$^{-1}$ (Pérez-Cruzado *et al.*, 2012), close to the 0.3 – 0.4 tC ha$^{-1}$ y$^{-1}$ of SOC accrual over 25 years given by our model. SOC accrual rates are then expected to decrease as the SOC stocks evolve towards a new steady-state.

Maps of SOC stocks are efficient tools to synthetize scientific results at the regional scale for stakeholders. They highlight on the one hand the areas where soil degradation would lead to the greatest release of $CO_2$ and, on the other hand, the areas with the highest potential for additional carbon storage. Despite the high uncertainties associated with regional-scale estimations of SOC stocks (Appendix Figures 2 and 3), our mean SOC stock values of 87 tC ha$^{-1}$ in the first 30 cm are in accordance with national-scale estimates that found SOC stocks of 75 – 100 tC ha$^{-1}$ in the North-East of France (Pellerin *et al.*, 2021).

The spatialization of current SOC stocks and additional storage capacity after 25 years at the regional scale was made possible by the uncommon abundance of soil profiles data and by the detailed pedological map available in the region of study. Such data-rich regions can serve as references for similar pedoclimatic zones. A further step would then be to intensify profile-scale data collection in other regions to provide reference values of SOC stocks and storage capacity in as many pedoclimatic zones as possible, in order to upscale this approach from the regional to the global scale (Barré *et al.*, 2017).

Increasing SOC stocks in soils has the potential to provide global benefits, but its successful implementation requires regional scale information on land use and soil type. While the impact of soil type on SOC accrual is only visible over long timescales, soil type remains a crucial factor to consider during land management decisions. Soil-type specific physico-chemical properties affect numerous soil functions such as water retention, resistance to erosion and nutrient cycling (Adhikari & Hartemink, 2016). These soil functions should be considered in addition to the SOC dynamics to choose management strategies adapted

to each soil type.

## 5 Conclusion

Informing stakeholders on soil management strategies to preserve and maximise existing soil organic carbon (SOC) stocks is a pressing concern to the scientific community. It is critical to communicate on the effects of soil type, depth and land-use on SOC accrual in soil over time periods compatible with the roadmap for C neutrality. This study explored how whole-profile

SOC accrual over decadal timescales differs from the SOC maximum additional storage capacity as estimated by current methods: according to texture (Hassink equation, 1997), or based on the top percentiles of regional stocks (data-driven), taking into consideration soil depth, soil type and land use.

The Hassink equation provided unrealistic profiles of SOC maximum additional storage capacity distribution at depth below 30 cm, as the equation only accounts for soil texture and does not consider the biotic controls on C inputs and SOC

decomposition rates. Depth-dependent profiles of maximum SOC stocks estimated from the data-driven approach were more in line with what is known of root distribution and therefore C input distribution with depth.

The estimation of SOC maximum additional storage capacity can be misleading if it is not estimated over timescales relevant to stakeholders. The SOC accrual modelled over 25 years in a scenario of high C inputs was five times lower compared to the SOC maximum additional storage capacity, which can only be reached over millennial timescales. We note a greater contrast

of SOC mean residence times at depth, which invites further investigation: while a fraction of the new C inputs added to the deep soil horizons can remain stable over millennial timescales, the majority is mineralized within two decades. Simulating a rise in temperatures of 1.3°C over 25 years following RCP8.5 attenuated SOC accrual by 10 to 50%.

The effect of soil type on SOC mineralization rates was not visible over the decadal timescales considered. However, soil type plays an important role on the spatial repartition of the current SOC stocks that need to be preserved. Studies of SOC stocks

and storage capacities should be complemented by more holistic explorations of soil functioning and ecosystem services which incorporate pedological knowledge.

This study provided a set of maps that give a complete picture of the issues related to carbon storage in soils (carbon stocks, additional storage capacities, and potentials for SOC accrual over decadal timescales). Such maps have the potential to facilitate communication with land planners and stakeholders by highlighting areas most worthy to preserve, and where carbon storage

practices are likely to be the most efficient over decadal timescales. The efficacy of such maps as decision support tools should be explored via collaboration projects with stakeholders.

**Author contributions**

All authors have given approval to the final version of the manuscript. The manuscript was written through contributions of all authors as follows:

CHIROL Clémentine: Conceptualization; data analysis; SOC model development; original draft

SÉRÉ Geoffroy: Conceptualization; writing-review & editing

REDON Paul-Olivier: data provider; writing-review & editing

CHENU Claire: Conceptualization; writing-review & editing

DERRIEN Delphine: Conceptualization; SOC model advice and improvement; writing-review & editing



**Declaration of competing interest**

The authors declare that they have no known competing financial interests or personal relationships that could have appeared to influence the work reported in this paper.

**Acknowledgements**

We received funding from the ANDRA, DEEPSURF and LUE for this project. We would like to thank Dr. Catherine Galy and Dr. Paul-Olivier Redon (ANDRA) for providing datasets and information on the OPE study area; Line Boulonne (RMQS) and Manuel Nicolas and Sébastien Macé (Renecofor) for providing datasets; and Dr. Laurent Saint-André and Dr. Marie-Pierre Turpault for their advice on the project.

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

**Appendices**

**Appendix 1: Details of model functioning**

A depth-dependent SOC dynamic model using multilayer soil modules was built to establish the time needed to reach different levels of carbon storage in the soil. SOC is allocated to three boxes (fast, slow, stable) corresponding to different SOC mineralization rates defined by Balesdent *et al.* (2018) based on a meta-analysis of changes in stable carbon isotope signatures at 55 grassland, forest and cropland sites, in the tropical zone. The mineralization rates were obtained using a $C_3/C_4$ approach, which is typically efficient to follow carbon dynamics over timescales ranging from one to one thousand years. Compared to the $^{14}C$ method, which covers timescales of several thousand years, the $C_3/C_4$ approach is relevant for land planning by exploring the impact of land use change on SOC dynamics (Verma *et al.*, 2017).

The mineralization factors associated with each box were then corrected for temperate soils using correction factors defined for the AMG model to account for the difference in environmental conditions (temperature and humidity) between tropical and temperate, but also to account for the differences in pH, clay content and $CaCO_3$ between soil types. The correction factors linked to temperature and humidity were based on Mary *et al.* (1999). The correction factors linked to pH, clay content and $CaCO_3$ were previously established by Clivot *et al.* (2017) based on the monitoring of N mineralization in 65 bare fallow soils representative of arable cropping systems in France, over a depth up to 150cm. These corrections are in accordance with recommendations from Rasmussen *et al.* (2018), for whom SOM stabilization not only depends on clay content, but also on pH and exchangeable calcium for alkaline soils. The correction factors for the temperature (T), humidity (H), clay content (A), pH and $CaCO_3$, as used in the 2019 AMG model, were as follows:

- $fT = \dfrac{25}{1+(25-1) * e^{0.12*15} * e^{-0.12*\text{T}}}$  *[Appendix Equation 1]*



- $fH = \dfrac{1}{1+0.03*e^{-5.247*(P-PET)/1000}}$     *[Appendix Equation 2]*

- $fA = e^{-2.519*10^{-3}*\text{Clay}}$     *[Appendix Equation 3]*

- $fpH = e^{-0.112*(pH-8.5)^2}$     *[Appendix Equation 4]*

- $fCaCO_3 = \dfrac{1}{1+(1.5*10^{-3}*\text{CaCO3})}$     *[Appendix Equation 5]*

The total correction factor f = fT * fH * fA * fpH * fCaCO₃, was calculated for both the 55 tropical sites from Balesdent *et al.*
(2018) and for the temperate conditions in the OPE region of the study, $f_{BAL}$ and $f_{OPE}$ respectively. The corrected mineralization
factors $k1_{corr}$ and $k2_{corr}$ were obtained with the following equations:

- $k1_{corr}$ = k1 * fOPE / fBAL     *[Appendix Equation 6]*

- $k2_{corr}$ = k2 * fOPE / fBAL     *[Appendix Equation 7]*

For each soil type and land use, the initial carbon stocks every 10 cm ($C_{init}$) was again obtained by data interpolation with the
Jreich method (2018); they were distributed between the three boxes based on the depth-dependent allocation factors defined
by Balesdent *et al.* (2018) (a1 and a2), as follows:

- $C1_{init}(i) = C_{init}(i) * a1$     *[Appendix Equation 8]*

- $C2_{init}(i) = C_{init}(i) * a2$     *[Appendix Equation 9]*

- $C3_{init}(i) = C_{init}(i) * (1-(a1+a2))$     *[Appendix Equation 10]*

The incorporated soil carbon inputs at each depth i and timestep t were added as follows:

- $C1_{in}(t,i) = INPUT(i) * \alpha(i)$     *[Appendix Equation 11]*

- $C2_{in}(t,i) = INPUT(i) * (1-\alpha(i))$     *[Appendix Equation 12]*

with $\alpha(i) = \dfrac{\frac{a1*k1corr}{a2*k2corr}}{1+(\frac{a1*k1corr}{a2*k2corr})}$     *[Appendix Equation 13]*

The outputs at each timestep were a function of the carbon stock at timestep t and of the corrected mineralization factors at
each depth i, as follows:

- $C1_{out}(t,i)=C1(t,i) * ( e^{-k1_{corr}(i)*\text{timestep}} - 1)$     *[Appendix Equation 14]*

- $C2_{out}(t,i)=C2(t,i) * ( e^{-k2_{corr}(i)*\text{timestep}} - 1)$     *[Appendix Equation 15]*





The change in soil carbon stock at each depth i between t and t+1 was defined as follows:

- $dC1(t,i)=C1_{out}(t,i) + C1_{in}(t,i)$                    *[Appendix Equation 16]*

- $dC2(t,i)=C1_{out}(t,i) + C2_{in}(t,i)$                    *[Appendix Equation 17]*

The soil carbon stocks at t+1 were therefore defined as:

- $C1(t+1,i) = C1(t,i) + dC1(t,i)$                    *[Appendix Equation 18]*

- $C2(t+1,i) = C2(t,i) + dC2(t,i)$                    *[Appendix Equation 19]*

The corrected mineralization rates also led to the definition of carbon mean residence times as a function of depth for each soil type (MRT = 1/k, see Appendix Table 2). SOC mean residence times at the steady state depend on the physico-chemical properties of the studied soil types: in our study site, they range from 50 – 100 years in the topsoil and from 145 – 453 years underneath.

The model was initialized under the assumption that the carbon stocks calculated at the different depths in 2018 were at steady state. This assumption is justified on average by a land occupation map from 1830 showing limited changes in land use over the past 200 years (Dupouey *et al.*, 2008). Inversing the model at the steady state yielded the vertical repartition of yearly C inputs necessary to keep the input and output fluxes equal across the full profile (Derrien & Amelung, see Appendix table 2 and Appendix Figure 1). We defined $INPUT_{eq}$ the repartition of incorporated C inputs every 10 cm at the steady state, as 745   follows:

- $C1_{eq}(i) = INPUT(i) * \frac{\alpha(i)}{k1_{corr}}$                    *[Appendix Equation 20]*

- $C2_{eq}(i) = INPUT(i) * \frac{(1-\alpha)}{k2_{corr}}$                    *[Appendix Equation 21]*

- $INPUT_{eq}(i) = \frac{C1_{init}(i) + C2_{init}(i)}{\frac{\alpha}{k1_{corr}}+\frac{(1-\alpha)}{k2_{corr}}}$                    *[Appendix Equation 22]*

This estimate of the yearly inputs did not distinguish between surface inputs and inputs by the root systems. The model further 750   assumed that there was no vertical redistribution of SOC between the layers following this initial allocation (Balesdent *et al.*, 2018). Then, the allocation and mineralization rates of these inputs were used at each depth layer to infer the mean residence time of the C inputs per land use: this second definition of the mean residence time depends on both the physico-chemical properties of the soil and on the vertical repartition of inputs.



**Appendix Table 1: List of soil properties collected at each soil profile and their measurement protocol**

| Study type | Soil Property | Unit | Method |
|---|---|---|---|
| Field observation | Slope | % | In situ operator's assessment |
| | Soil depth | Cm | In situ operator's assessment |
| | Horizon Textural Class | Type | In situ operator's assessment completed by NF X 31-107 |
| | Horizon Compacity | Type | knife test (ISO 25177: 2008) |
| | Horizon Rock Fragment Content | % | In situ operator's assessment |
| | Horizon Hydromorphic Features | Type | In situ operator's assessment |
| Lab Agronomical Analysis | Horizon pH | - | NF ISO 10390 |
| | Horizon OM | g/kg | NF ISO 10694 |
| | Horizon CaCO3 | g/kg | NF ISO 10693 |

**Appendix Table 2: List of descriptors used to plot the SOC content curves for each soil type and land use: $\Omega_1$ the SOC content of the soil type at maximal depth, $\Omega_2$ the SOC content at the surface, and $\Omega_3$ the depth at half maximum of the SOC content (based on Mathieu *et al.* (2015) and Jreich (2018))**

| Land use | Soil type (WRB) | Soil type (RPF) | $\Omega_1$ Bottom SOC (g/kg) | $\Omega_2$ Top SOC (g/kg) | $\Omega_3$ Depth at half maximum of the carbon content (cm) |
|---|---|---|---|---|---|
| Cropland | Calcaric rendzic leptosol | Rendosol | 17 | 31 | 17 |
| Forest | Calcaric rendzic leptosol | Rendosol | 22 | 74 | 16 |
| Grassland | Calcaric rendzic leptosol | Rendosol | 12 | 53 | 15 |
| Cropland | Calcaric cambisol | Calcosol | 14 | 33 | 21 |
| Forest | Calcaric cambisol | Calcosol | 17 | 62 | 18 |
| Grassland | Calcaric cambisol | Calcosol | 14 | 54 | 15 |
| Cropland | Hypereutric epileptic cambisol | Rendisol | 19 | 38 | 13 |





| | | | | | | |
|---|---|---|---|---|---|---|
| Forest | Hypereutric epileptic cambisol | Rendisol | 16 | 60 | 12 |
| Cropland | Hypereutric cambisol | Calcisol | 10 | 24 | 17 |
| Forest | Hypereutric cambisol | Calcisol | 22 | 64 | 21 |
| Grassland | Hypereutric cambisol | Calcisol | 14 | 54 | 15 |
| Cropland | Eutric cambisol | Brunisol | 8 | 18 | 21 |
| Forest | Eutric cambisol | Brunisol | 8 | 45 | 16 |
| Grassland | Eutric cambisol | Brunisol | 5 | 23 | 21 |
| Forest | Dystric cambisol | Alocrisol | 4 | 31 | 15 |
| Cropland | Stagnosol | Rédoxisol | 10 | 21 | 19 |
| Forest | Stagnosol | Rédoxisol | 9 | 46 | 17 |
| Grassland | Stagnosol | Rédoxisol | 9 | 40 | 14 |
| Cropland | Gleysol | Réductisol | 16 | 26 | 16 |
| Grassland | Gleysol | Réductisol | 21 | 68 | 18 |


**Appendix Table 3: Details of the SOC average mean residence time (MRT = 1/k) in the fast pool (MRT$_1$ = 1/k$_1$) and in the slow pool (MRT$_2$ = 1/k$_2$), represented in years as a function of depth for each soil type, using parameters from Balesdent *et al.* (2018), with correction factors from the AMG model for the temperature, P/PET, pH, clay content and CaCO$_3$. The value α represents the proportion of new carbon inputs that is allocated to the fast carbon pool (see Appendix Equation 13). Soil 1: Calcaric rendzic leptosol,**
**soil 2: Calcaric cambisol; soil 3: Hypereutric epileptic cambisol; soil 4: Hypereutric cambisol; soil 5: Eutric cambisol; soil 6: Dystric cambisol; soil 7: Stagnosol; soil 8: Gleysol.**

| Depth (cm) | α | Average MRT (y) | | | | | | | | | | | | | | |
|---|---|---|---|---|---|---|---|---|---|---|---|---|---|---|---|---|
| | | Soil 1 | | Soil 2 | | Soil 3 | | Soil 4 | | Soil 5 | | Soil 6 | | Soil 7 | | Soil 8 | |
| | | MRT1 | MRT2 | MRT1 | MRT2 | MRT1 | MRT2 | MRT1 | MRT2 | MRT1 | MRT2 | MRT1 | MRT2 | MRT1 | MRT2 | MRT1 | MRT2 |
| 0 – 10 | 0.98 | 22 | 628 | 20 | 563 | 22 | 630 | 20 | 566 | 26 | 742 | 38 | 1100 | 23 | 664 | 17 | 477 |
| 10 – 20 | 0.92 | 31 | 777 | 27 | 696 | 31 | 779 | 28 | 701 | 36 | 918 | 37 | 948 | 32 | 822 | 23 | 591 |
| 20 – 30 | 0.86 | 13 | 643 | 14 | 676 | 26 | 1284 | 23 | 1121 | 29 | 1422 | 24 | 1190 | 21 | 1031 | 15 | 741 |
| 30 – 40 | 0.86 | 13 | 863 | 13 | 908 | 25 | 1724 | 22 | 1505 | 28 | 1910 | 25 | 1727 | 14 | 977 | 13 | 892 |
| 40 – 50 | 0.88 | | | 11 | 1013 | 26 | 2321 | 23 | 2026 | 29 | 2571 | 22 | 1943 | 15 | 1315 | 14 | 1200 |
| 50 – 60 | 0.90 | | | 13 | 1317 | | | 44 | 4654 | 30 | 3198 | 24 | 2526 | 16 | 1710 | 15 | 1561 |
| 60 – 70 | 0.91 | | | | | | | 47 | 5171 | 32 | 3553 | 25 | 2807 | 17 | 1900 | 16 | 1734 |
| 70 – 80 | 0.91 | | | | | | | 42 | 5817 | 29 | 3997 | 23 | 3158 | 14 | 1948 | 14 | 1951 |
| 80 – 90 | 0.91 | | | | | | | 37 | 5817 | 26 | 3997 | 20 | 3158 | 12 | 1948 | 11 | 1744 |
| 90 – 100 | 0.92 | | | | | | | | | | | 22 | 3501 | 12 | 1948 | 11 | 1744 |
| Average MRT above 30 cm | | 62 | | 57 | | 69 | | 62 | | 81 | | 100 | | 70 | | 50 | |
| Average MRT below 30 cm | | 145 | | 155 | | 309 | | 453 | | 418 | | 384 | | 226 | | 206 | |





**Appendix Table 4: Vertical repartition in % of yearly C inputs at the steady state (stationary C inputs) for each soil type, land use and depth layer every 10 cm. The bottom of the table provides the total inputs in tC ha$^{-1}$ y$^{-1}$ needed to stay at the steady state, or to reach the maximum SOC stocks estimated by the 75$^{th}$ percentile data-driven method. C = Cropland; F = Forest; G = Grassland.**

| Depth (cm) | Calcaric rendzic leptosol | | | Calcaric cambisol | | | Hypereutric epileptic cambisol | | Hypereutric cambisol | | | Eutric cambisol | | | Dystric cambisol | Stagnosol | | | Gleysol | |
|---|---|---|---|---|---|---|---|---|---|---|---|---|---|---|---|---|---|---|---|---|
| | C | F | G | C | F | G | C | F | C | F | G | C | F | G | F | C | F | G | C | G |
| 0 | 0.672 | 0.709 | 0.737 | 0.701 | 0.700 | 0.764 | 0.739 | 0.772 | 0.705 | 0.699 | 0.743 | 0.668 | 0.738 | 0.760 | 0.629 | 0.499 | 0.651 | 0.630 | 0.489 | 0.606 |
| 10 | 0.173 | 0.167 | 0.158 | 0.148 | 0.155 | 0.128 | 0.184 | 0.172 | 0.164 | 0.171 | 0.154 | 0.160 | 0.157 | 0.151 | 0.174 | 0.130 | 0.149 | 0.142 | 0.108 | 0.141 |
| 20 | 0.116 | 0.096 | 0.082 | 0.082 | 0.083 | 0.060 | 0.045 | 0.034 | 0.054 | 0.055 | 0.046 | 0.061 | 0.049 | 0.045 | 0.061 | 0.066 | 0.060 | 0.060 | 0.056 | 0.061 |
| 30 | 0.039 | 0.028 | 0.023 | 0.033 | 0.031 | 0.023 | 0.030 | 0.020 | 0.032 | 0.032 | 0.025 | 0.037 | 0.023 | 0.020 | 0.025 | 0.058 | 0.041 | 0.044 | 0.043 | 0.037 |
| 40 | | | | 0.020 | 0.018 | 0.014 | 0.003 | 0.002 | 0.024 | 0.023 | 0.018 | 0.023 | 0.012 | 0.010 | 0.019 | 0.047 | 0.026 | 0.030 | 0.038 | 0.026 |
| 50 | | | | 0.015 | 0.013 | 0.011 | | | 0.008 | 0.007 | 0.005 | 0.017 | 0.008 | 0.005 | 0.013 | 0.038 | 0.017 | 0.021 | 0.033 | 0.019 |
| 60 | | | | | | | | | 0.007 | 0.006 | 0.005 | 0.015 | 0.006 | 0.004 | 0.011 | 0.034 | 0.013 | 0.017 | 0.030 | 0.016 |
| 70 | | | | | | | | | 0.006 | 0.005 | 0.004 | 0.013 | 0.005 | 0.003 | 0.009 | 0.030 | 0.010 | 0.014 | 0.026 | 0.013 |
| 80 | | | | | | | | | 0.002 | 0.002 | 0.001 | 0.006 | 0.002 | 0.001 | 0.009 | 0.029 | 0.009 | 0.013 | 0.028 | 0.013 |
| 90 | | | | | | | | | | | | | | | 0.008 | 0.027 | 0.009 | 0.012 | 0.027 | 0.013 |
| 100 | | | | | | | | | | | | | | | 0.009 | 0.030 | 0.010 | 0.013 | 0.031 | 0.014 |
| 110 | | | | | | | | | | | | | | | 0.009 | 0.012 | 0.004 | 0.005 | 0.031 | 0.014 |
| 120 | | | | | | | | | | | | | | | 0.009 | | | | 0.031 | 0.014 |
| 130 | | | | | | | | | | | | | | | 0.009 | | | | 0.028 | 0.013 |
| 140 | | | | | | | | | | | | | | | 0.009 | | | | | |
| Total inputs to stay at the steady state (tC ha$^{-1}$ y$^{-1}$) | 1.34 | 2.75 | 1.91 | 1.84 | 2.83 | 2.73 | 1.47 | 1.97 | 1.36 | 2.26 | 2.51 | 0.98 | 2.02 | 1.19 | 1.03 | 1.50 | 2.33 | 1.92 | 2.79 | 4.59 |
| Total inputs to reach Max SOC (tC ha$^{-1}$ y$^{-1}$) | 3.15 | 3.61 | 2.20 | 3.14 | 2.26 | 1.44 | 3.22 | 5.99 | 3.15 | 3.61 | 2.20 | 3.14 | 2.26 | 1.44 | 3.22 | 5.99 | 3.15 | 3.61 | 2.20 | 3.14 |

**Appendix Table 5: SOC stocks and maximum storage capacity above and under 30 cm (under 30 cm represented in bold)**

| | Median SOC stocks in 2018 (tC ha$^{-1}$) | Maximum stocks (75$^{th}$ percentile) (tC ha$^{-1}$) | SOC maximum additional storage capacity (tC ha$^{-1}$) | |
|---|---|---|---|---|
| | | | Based on maximum | Based on maximum stock under all land uses |



| | | | | | | stock under croplands | | | |
|---|---|---|---|---|---|---|---|---|---|
| | Cropland | Grassland | Forest | Cropland | All land uses | Cropland | Cropland | Grassland | Forest |
| Calcaric Rendzic Leptosol | 70 **8** | 95 **6** | 138 **11** | 89 **11** | 155 **12** | 19 **3** | 85 **5** | 60 **6** | 17 **2** |
| Calcaric Cambisol | 81 **19** | 114 **20** | 123 **24** | 133 **30** | 155 **36** | 52 **11** | 75 **17** | 41 **16** | 32 **12** |
| Hypereutric Epileptic Cambisol | 78 **14** | | 97 **10** | 98 **18** | 112 **17** | 20 **4** | 34 **3** | | 15 **7** |
| Hypereutric Cambisol | 63 **40** | 113 **54** | 104 **56** | 71 **54** | 142 **86** | 8 **14** | 78 **46** | 28 **32** | 38 **30** |
| Eutric Cambisol | 59 **43** | 71 **19** | 119 **38** | 79 **70** | 130 **64** | 20 **27** | 71 **22** | 59 **45** | 11 **27** |
| Dystric Cambisol | | | 76 **44** | | 101 **68** | | | | 25 **24** |
| Stagnosol | 64 **101** | 92 **69** | 76 **44** | 79 **164** | 142 **143** | 15 **63** | 76 **42** | 50 **74** | 28 **85** |
| Gleysol | 78 **202** | 156 **177** | 114 **58** | 91 **269** | 187 **289** | 13 **67** | 110 **87** | 32 **111** | |



**Appendix Table 6: simulated SOC accrual in tC ha$^{-1}$ in the different soil types and land uses (C= cropland, F=forest, G=grassland) after 1, 10, 50, 200, 1000 and 5000 years of model run under a scenario of additional inputs of 0.5 tC ha$^{-1}$ y$^{-1}$ under forests, 1.0 tC ha$^{-1}$ y$^{-1}$ under grasslands and 1.5 tC ha$^{-1}$ y$^{-1}$ under croplands.**

| Years | Calcaric rendzic leptosol | | | Calcaric cambisol | | | Hypereutric epileptic cambisol | | Hypereutric cambisol | | | Eutric cambisol | | | Dystric Cambisol | Stagnosol | | | Gleysol | |
|---|---|---|---|---|---|---|---|---|---|---|---|---|---|---|---|---|---|---|---|---|
| | C | G | F | C | G | F | C | F | C | G | F | C | G | F | F | C | G | F | C | G |
| 1 | 1.5 | 1.0 | 0.5 | 1.5 | 1.0 | 0.5 | 1.5 | 0.5 | 1.5 | 1.0 | 0.5 | 1.5 | 1.0 | 0.5 | 0.5 | 1.5 | 1.0 | 0.5 | 1.5 | 1.1 |
| 10 | 12.6 | 8.6 | 4.6 | 12.5 | 8.6 | 4.6 | 12.9 | 4.6 | 12.6 | 8.7 | 4.5 | 13.1 | 8.8 | 4.6 | 4.6 | 12.8 | 8.8 | 4.6 | 12.3 | 8.9 |
| **25** | **23.7** | **16.2** | **8.7** | **22.9** | **15.8** | **8.4** | **24.5** | **8.6** | **23.7** | **16.2** | **8.5** | **26.0** | **17.3** | **9.1** | **9.4** | **23.6** | **16.4** | **8.7** | **21.3** | **15.5** |
| 50 | 32.6 | 22.2 | 11.9 | 30.6 | 21.0 | 11.2 | 34.3 | 11.9 | 32.2 | 21.9 | 11.5 | 37.5 | 24.8 | 13.0 | 14.4 | 33.1 | 23.1 | 12.2 | 27.6 | 20.1 |
| 100 | 39.1 | 26.3 | 14.0 | 36.0 | 24.4 | 13.0 | 40.9 | 14.0 | 38.2 | 25.7 | 13.5 | 46.5 | 30.3 | 15.9 | 18.8 | 40.6 | 27.9 | 14.6 | 32.7 | 23.3 |
| 200 | 45.5 | 30.0 | 16.0 | 41.8 | 27.8 | 14.9 | 46.7 | 15.7 | 44.2 | 29.4 | 15.4 | 54.0 | 34.5 | 18.0 | 22.1 | 49.1 | 32.7 | 16.8 | 40.1 | 27.6 |
| 5000 | 84.5 | 52.1 | 27.4 | 78.9 | 48.4 | 26.4 | 92.4 | 27.6 | 98.5 | 60.1 | 32.1 | 133.8 | 69.6 | 36.2 | 50.0 | 142.3 | 77.9 | 36.1 | 118.5 | 64.1 |
| | SOC accrual after 25 years under temperature increase of 1.0 °C by 2050 (RCP 4.5 scenario) | | | | | | | | | | | | | | | | | | | |
| **25** | **21.5** | **13.6** | **5.5** | **20.2** | **12.5** | **5.3** | **22.1** | **6.2** | **21.4** | **13.0** | **5.8** | **24.1** | **15.5** | **6.5** | **8.0** | **21.3** | **13.8** | **5.9** | **17.8** | **10.5** |
| | SOC accrual after 25 years under temperature increase of 1.3 °C by 2050 (RCP 8.5 scenario) | | | | | | | | | | | | | | | | | | | |



| 25 | 20.8 | 12.9 | 4.5 | 19.4 | 11.4 | 4.3 | 21.4 | 5.5 | 20.7 | 12.1 | 5.0 | 23.5 | 14.9 | 5.8 | 7.5 | 20.5 | 13.0 | 5.1 | 16.7 | 9.0 |


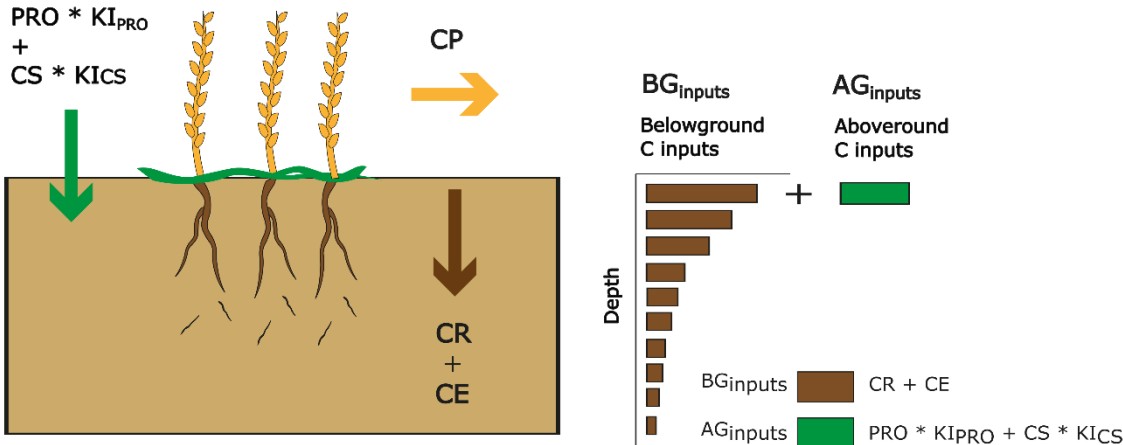

CE = extra-root C (C content in roots * 0.65)
CP = C product (crop yield * C content in plant parts)
CR = C roots ( CP * Root:Shoot ratio / Harvest Index )
CS = C straw ( CP * ( 1 - Harvest Index ) / Harvest Index )
$KI_{CS}$ = coefficient of incorporation for the straw (0.1, Girard et al., 2011)
$KI_{PRO}$ = coefficient of incorporation for the organic amendments (0.3, Girard et al., 2011)
PRO = Organic amendments from manure * organic matter content in manure * C content in organic matter

C content in plant parts = 0.45 (Bolinder et al., 2007)
Harvest Index = 0.4 (Bolinder et al., 2007)
Organic matter content in manure = 0.57 (INRAE - MAFOR)
Root:Shoot ratio = 0.1 in croplands (Jackson et al., 1996)

**Appendix Figure 1: Estimation of the current incorporated C inputs in croplands via a yield-based allocation coefficients method from Bolinder *et al.* (2017) using agricultural yield and amendment values based on compiled reports from 2010-2019 in the region of study. The allocation coefficients were derived from the literature (harvest index and carbon content in plant parts from Bolinder *et al.* (2007), organic matter content in manure from Houot *et al.* (2014), root:shoot ratios in croplands from Jackson *et al.* (1996), incorporation coefficients form Girard *et al.* (2011)). Estimated C inputs in the croplands in the region of study are 1.4tC ha$^{-1}$ y$^{-1}$, with a mean winter wheat yield value of 5.53 tDM ha$^{-1}$ y$^{-1}$ and an amendment value of 2.13 tDM ha$^{-1}$ y$^{-1}$. The average C inputs at the steady state obtained via model inversion in the croplands of the region of study, weighted by the proportion of each soil type in the cropland areas, amount to 1.7 tC ha$^{-1}$ y$^{-1}$.**




# Local SOC stock variability due to the non-explicit repartition of soil types in each cartographic unit

[Map showing Local variability (tC/ha) with legend: 0 - 10, 11 - 20, 21 - 30, 31 - 40, 41 - 50, 51 - 60, 61 - 70; zones 1 and 2 marked; scale bar 0 1 2 4 6 8 Kilometers]


**Appendix Figure 2: Local uncertainty of SOC linked to the non-explicit repartition of soil types within the cartographic units. As an example, in zone 1, which is under forest, the represented soil types are 80% Eutric cambisol (157 tC ha$^{-1}$) and 20% Stagnosol (172 tC ha$^{-1}$). In zone 2, which is under grassland, the represented soil types are 80% Stagnosol (161 tC ha$^{-1}$) and 20% Gleysol (333 tC ha$^{-1}$). For this reason, the local variability of SOC stocks is higher in zone 2 than zone 1.**








**Appendix Figure 3: SOC stocks and maximum SOC additional storage capacity, with lower and upper confidence intervals as estimated by the bootstrap method. The SOC stock in the region of study ranges from 2.4 – 5.3 MtC and the maximum SOC additional storage capacity 1.2 - 4.1 MtC.**



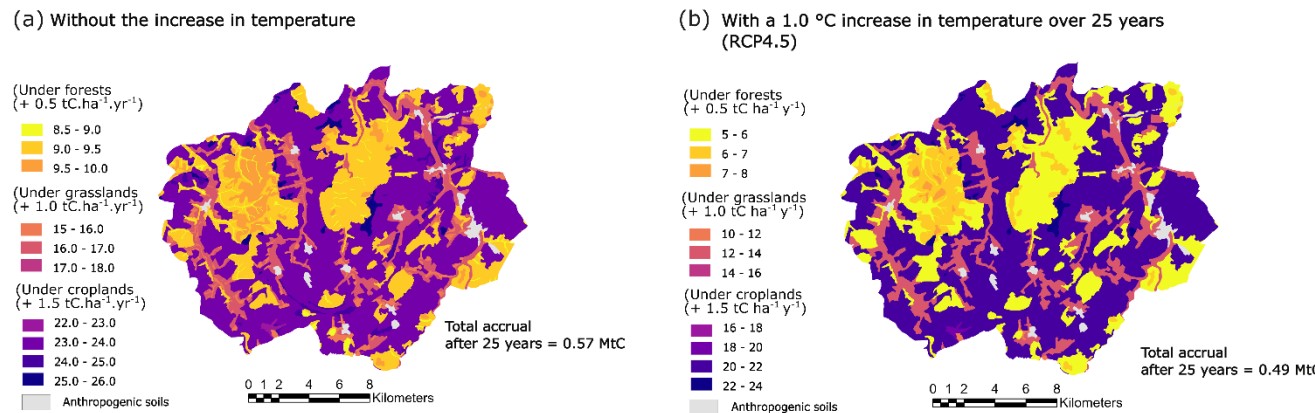

SOC accrual after 25 years (tC ha⁻¹) under a scenario of additional C inputs dependent on land use

(a) Without the increase in temperature

(b) With a 1.0 °C increase in temperature over 25 years (RCP4.5)

**Appendix Figure 4: SOC accrual after 25 years under a scenario of additional C inputs dependent on land use, (a) with temperatures staying at their 2018 level, and (b) with a 1.0 °C increase in temperature over 25 years, increasing the C mineralization rates according to the correction factors of the AMG model. The attenuation in SOC accrual due to increased mineralization rates is (0.49 – 0.57) / 0.57 = 14%. The 1.0 °C increase in temperature was obtained from model simulations of mean annual temperatures by the Meteo France ALADIN63_CNRM-CM5 model under scenario RCP4.5, within an 8 km radius area around Bure (55087), comparing the year intervals 2046-2055 and 2009-2019. Source: Drias, données Météo-France, CERFACS, IPSL.**