# Peer review of "Depth-dependence of soil organic carbon additional storage capacity in different soil types by the 2050 target for carbon neutrality"

_EGUsphere, 2024_

## Author Comment (AC1)

General Comments
In this manuscript, Chirol et al. combine direct measurements of deep soil cores with process modelling to investigate the potential for soils in a region of North-eastern France to accrue SOC over 25 years under a scenario of increased C inputs. While the topic is timely and important, I found the logic of the paper hard to follow and the methods to be a bit convoluted. In my opinion, the manuscript needs a clearer storyline throughout, from the hypothesis or research question, to the approach, and through to the discussion. Why was this particular approach chosen, what are the strengths, and what exact gaps does it fill? While elements of this are there, it was really not clear enough to follow easily and I was left with many questions as I read it. The methods were presented in a bit of a circular way and were not well-justified, especially around the determination of potential SOC accrual and the input rates used (see below), and the AGM model, which is integral to the paper, is not well-described in the main text at all. I have four major recommendations for the manuscript:

**Thank you for the thorough and constructive criticism of our work. In our responses below and in the revised version of the manuscript, we will attempt to clarify our approach, and how it fulfils the objectives of the research.**

1. More clearly present the logic and reasoning behind comparing two approaches (Hassink vs. Chen et al.) for estimating potential SOC storage. First, there needs to be clearer description of and distinction between the idea of fine fraction saturation vs. maximum SOC accrual, which are two very different concepts (though saturation may be a constraint on maximum SOC accrual), and which relate directly to the conclusions and the implications of the work. I have several specific comments on this below. Second, how do these ideas and assumptions relate to the conclusions drawn from them (see #4)? It is critical to be careful in how one presents and interprets these "maximum values".

   **We will clarify our description of the concepts of saturation and maximal SOC accrual, insisting on the fact that they do not provide the same variable and are not comparable from a mechanism perspective – however the two concepts are used to achieve the same objective: to provide quantitative targets for SOC sequestration for stakeholders.**

   **Considering that the main objective of the paper is to estimate and map realistic targets for SOC sequestration within decadal timescales, accounting for soil type and depth, it appears from the referee comments that including considerations about the fine fraction saturation approach could mislead the reader.**

   **Consequently, we propose to state in the introduction that the fine fraction saturation is not a pertinent constraint on additional SOC storage capacity at depth contrary to the maximal SOC accrual approach - Additionnal arguments supporting this have also been very recently provided by Peoplau et al (2024, GCB) – and to refocus the paper on our modelling exercise to investigate the kinetics of C stock build up under different scenarios of additional C inputs.**

   **In a practical way, references to POM and MAOM will be removed from the introduction (Line 39-49). Line 50-54 and 86-89 will be removed. The Hassink approach will be removed from the Methods section 2.2.2 (Line 165-180). In the Results section, the curves derived from the Hassink equation will be removed**

**from Figure 3b, as well as the text related to those curves. In the Discussion section, the lines 361-383 will be removed from the main text.**

**Our exploration of depth-dependent Hassink curves will be moved to a Supplementary material. We deem this valuable because the high level of undersaturation in the deeper horizons can give the false impression that their SOC stock can be easily increased. Explicitly showing the depth-dependent saturation curve makes it obvious that undersaturation is not pertinent to define SOC storage capacity at depth.**

2. Provide better rationale of the approach in terms of the scenarios modeled. The input rates were chosen based on what the model requires to maintain the theoretical upper limits of SOC accrual by land use at steady state, and then those input rates were simulated to see how much SOC accrued after 25 years. Why not instead use the accrual rates necessary to achieve theoretical maximum SOC storage to inform ideal inputs and see if they are achievable, or use more realistic input rates (perhaps ranges) and see how far from the maximum SOC accrual those are after 25 years?

**We agree that this approach will clarify the storyline of our research and thank the reviewer for the suggestion.**

**In total, three different C input scenarios will be modelled:**

1) **An initial conditions input regime corresponding to the yearly C inputs necessary to maintain the initial SOC stocks in each soil type and land use: there is no SOC accrual in this case.**
2) **A realistic increased input regime of +0.5 tC/ha/y in forests, +1.0 tC/ha/y in grasslands, and +1.5 tC/ha/y in croplands. We used the following sources:**
   a. **Initial C inputs in forests range within 1.6-2.8 tC/ha/y according to measurements carried out in the Renecofor network in the region of study, assuming 50% mineralisation of above ground input in the forest floor. Additional inputs from harvest residues after thinning range within 0.5 – 2 tC/ha/y, leading to a total realistic range of 1.6 – 4.8 tC/ha/y.**
   b. **In grasslands, annual inputs to the soil range within 1.18 – 5.2 tC/ha/y according to studies from Australia and Western Europe (methods used: RothC inverse modelling, allometric equations using yield data, expert opinion) (Martin et al., 2020)**
   c. **In croplands, annual inputs to the soil range within 1.8 – 6.8 tC/ha/y according to studies conducted worldwide (methods used: direct measurements, RothC inverse modelling, allometric equations using yield data, expert opinion) (Martin et al., 2020)**
   **These values are compatible with what the model requires to maintain the theoretical maximum SOC stock at steady state, which confirms the robustness of our approach.**
3) **An extreme input regime necessary to reach the theoretical maximum SOC stock within 25 years**

**We have concatenated the results from these three scenarii in the table below, which will replace Table 2 in the new version of the paper:**

**Table 2: Initial SOC stocks, C input regimes to the soil considered in this study, theoretical maximum SOC stocks based on the 75[th] percentile of our regional dataset, and SOC stock after 25 years under a realistic scenario of C inputs, for each soil type and land use. Realistic range of annual C inputs to the soil is 1.8 – 6.8 tC/ha/y for croplands (Martin et al., 2020), 1.18 – 5.2 tC/ha/y for grasslands (Martin et al., 2020), and 1.6 – 4.8 tC/ha/y for forests according to measurements made in the region of study.**

| Depth (cm) | Calcaric rendzic leptosol | | | Calcaric cambisol | | | Hypereutric epileptic cambisol | | Hypereutric cambisol | | | Eutric cambisol | | | Dystric cambisol | Stagnosol | | | Gleysol | |
|---|---|---|---|---|---|---|---|---|---|---|---|---|---|---|---|---|---|---|---|---|
| | C | G | F | C | G | F | C | F | C | G | F | C | G | F | F | C | G | F | C | G |
| Initial SOC stock (tC ha$^{-1}$) | 78 (48-115) | 101 (84-138) | 149 (97-183) | 100 (58-133) | 134 (66-183) | 148 (104-184) | 92 (49-129) | 106 (76-121) | 103 (62-137) | 167 (125-255) | 160 (92-204) | 102 (59-144) | 90 (66-115) | 157 (71-190) | 120 (76-198) | 166 (101-237) | 161 (108-279) | 172 (121-249) | 279 (154-417) | 333 ((252-466) |
| Initial input regime (tC ha$^{-1}$ y$^{-1}$) | 1.3 | 1.9 | 2.7 | 1.8 | 2.7 | 2.8 | 1.5 | 2.0 | 1.4 | 2.5 | 2.3 | 1.0 | 1.2 | 2.0 | 1.0 | 1.5 | 1.9 | 2.3 | 2.8 | 4.6 |
| Realistic increased input regime (tC ha$^{-1}$ y$^{-1}$) | 2,8 | 2,9 | 3,2 | 3,3 | 3,7 | 3,3 | 3 | 2,5 | 2,9 | 3,5 | 2,8 | 2,5 | 2,2 | 2,5 | 1,5 | 3 | 2,9 | 2,8 | 4,3 | 5,1 |
| Extreme input regime (tC ha$^{-1}$ y$^{-1}$) | 7.0 | 6.0 | 3.8 | 7.9 | 6.5 | 5.7 | 3.8 | 3.4 | 9.5 | 6.4 | 6.7 | 6.4 | 7.4 | 4.2 | 3.7 | 9.2 | 9.8 | 9.4 | 17.3 | 14.9 |
| Theoretical maximum SOC stock (tC ha$^{-1}$) | 167 | | | 191 | | | 129 | | 228 | | | 194 | | | 169 | 285 | | | 476 | |
| SOC stock after 25 years under realistic increased input regime (tC ha$^{-1}$) | 102 | 118 | 157 | 123 | 150 | 156 | 117 | 115 | 127 | 183 | 168 | 128 | 107 | 166 | 129 | 189 | 177 | 181 | 301 | 349 |

3. Provide more detail on the AGM model, especially whether/how it represents saturation and temperature sensitivity of carbon pools and how those relate to the concepts of MAOM and POM mentioned early in the manuscript. Why is this an appropriate model for this use, and what are the implications of the model structure (and perhaps its limitations) for the results and conclusions?

**We apologize for not being clear enough in our definition of the model and will modify the introduction and Material and Method section (between Line 195-210) as well as Figure 2 to improve the model description, insisting on the following elements:**

**The model we applied is not the AGM model. Instead, it is a three-pool model with a fast cycling, a slow cycling pool and an inert pool. Dynamics pools are ruled by exponential kinetics. Pool size and turnover have been calibrated by**

Balesdent et al. (Nature, 2018) using a global database of 13C isotopes measured in multiple campaigns, principally over several decades. This is a crucial strength for our objective to estimate C accrual over 25 years. The model is also layered into 10 cm sections to account for variations in mean residence times with depth.

We corrected the model parameters calibrated at the global scale in Balesdent et al. (2018) to account for local conditions of temperature, humidity, pH, clay content and $CaCO_3$ content as recommended by Rasmussen et al. (2018). Here we used the equations of the AGM model, which likely explains the confusion.

In the model, SOC stock does not saturate, and is dependent on C inputs. The model does not intend either to represent measurable pools such as POM and MAOM.

The model does not account for vertical transfer, but Balesdent et al (2018) showed that 13C incorporation in subsoil after a change in vegetation is slow and affects only long-term carbon dynamics. It is therefore negligible at decadal timescales. Sierra et al (2024, GCB) also found that transport may only play a secondary role in the formation of soil carbon profiles according to simulation examples and measurements from carbon and radiocarbon profiles.

The model does not take into account the priming effect, even though it is expected to occur when C inputs to the soil increase. Priming is difficult to model because the processes involved are still poorly understood (Bernard et al., 2022). Current explorations of the priming effect use either mechanistic models centred on microbial processes (Schimel, 2023), or theoretical models fitted to laboratory experiments, which do not fit the scope of our study.

We propose to modify Figure 2 to better illustrate our modelling approach (see below). This version makes it clearer that there is no vertical redistribution in our model:

[Figure]

**Suggested New Figure 2: Summary of our approach: (a) estimation of initial and theoretical maximum SOC stocks from the measured data; (b) estimation of vertical repartition of C input for the different scenarios considered; (c) Functioning of the depth-dependent three-pool model (fast-cycling pool, slow-cycling pool, inert pool). a = allocation factor ; MRT = Mean Residence Time (in years). MRT values vary with depth as per Balesdent et al. (2018) and are corrected for temperature, humidity, pH, texture and CaCO₃ (see Methods).**

4. The discussion can be much clearer about the implications of the approaches used and dig more into how different assumptions end up informing potential strategies for climate mitigation. How does considering depth change our understanding of how we might manage soils for climate? What about methods of determining maximum potential C storage (and trajectories for reaching it)? Should we even base management ideas around these maxima, or are they infeasible to reach on relevant timescales?

**We will rework and develop the discussion section following your proposed plan, as follows (section titles and plan are not definitive):**

**Section 4.1: Implications of our approaches to estimate target SOC stocks and accrual rates**

> **-Compare the realistic increased inputs and the extreme inputs regimes to show that the latter is unrealistic. Confront our realistic increased inputs to the literature, improving on our previous attempt (Lines 458-469);**
>
> **-Discuss the effects of land use, soil type and the consideration of depth on SOC stocks, theoretical maximum SOC accrual, and on modelled C accrual rates (Effect of soil type versus land use = Lines 436-446 ; Effect of depth = Lines 407-415);**
>
> **-Briefly discuss the lack of consensus within the scientific community on the choice of percentile value to define theoretical maximum stocks. (Lines 388-404);**
>
> **-Mention the limitations of the model (no priming effect) (Lines 416 – 421 + our arguments in response to General Comment 3);**
>
> **-Discuss our approach for testing the effect of temperature increase on C accrual rates, and the limitations of these tests. (Lines 422-435);**

**Section 4.2: Implications for stakeholders (what level of C accrual is achievable after 25 years)**

> **-Summarize what is achievable after 25 years in the region of study (accounting or not for temperature increase), and how it differs from the theoretical maximum SOC stocks (Lines 449-457);**
>
> **-Highlight the importance of maps of initial SOC stocks and achievable SOC accrual within decadal timescales for stakeholder decision-making (maps of theoretical maximum SOC accrual appear less useful), improving on Lines 470 – 479);**

**-Discuss the difficulties in universally increasing C inputs to various lands, (especially at depth, and especially when taking into account socio-economic factors) using the ranges from the literature listed in response to General Comment 2;**

**-Discuss in what circumstances adding carbon to soils might not be the best solution, by mentioning other possible uses for biomass, and the necessity to analyse the life cycle of any biomass brought to the system (see Derrien 2023, section 4). Reminder of the importance of soils for ecosystem services other than carbon storage (Lines 480-485).**

Specific Comments

Lines 50-54: I find this hard to understand and perhaps misleading, as there are several issues that are being confounded, and depth is not quite relevant in the way it is said here. My reading of the Chen et al. (2018) paper was that their approach not only identified many soils that were undersaturated (as pointed out here), but also identified many soils that were "oversaturated" according to the Hassink equation. The latter is because the Hassink equation was developed for MAOM only and is a least-squares linear regression approach which results in many soils falling above the predicted saturation level and is precisely why boundary approaches (but still typically for MAOM only) have gained favor over the past decade (Feng et al., 2013). Further, given that the Hassink approach was not developed on whole soil profiles (top 10 cm only), applying it to a whole profile does not seem a relevant solution to the issue presented in lines 53-54. The actual issue at hand here is that these two methods are not compatible and should not be directly compared, as one pertains only to saturation of the fine fraction while the other looks at whole soil carbon stocks.
**Thank you for the comment, on which we agree. As mentioned in our response to General Comment 1, in line with your recommendations, we have decided to focus the paper on the theoretical maximum SOC accrual approach. This section will be deleted.**

Lines 55-65: While I agree that better understanding of subsoil C dynamics and its controls is important, it is doubtful whether saturation or storage capacity are important constraints on subsoil C accrual. This paragraph should better connect subsoils with the topic of saturation and ideally address why it may or may not be an important constraint on subsoil C accrual head-on. This point is very important to the justification, methods, and conclusions of the paper and can further clarify the aims and ideal application of the two contrasting methods (fine fraction saturation vs. whole C storage capacity). In general it should be pointed out that a saturation approach is not ideal for estimating the target of subsoil C accrual because fine fraction saturation is unlikely to be a relevant constraint; this approach may therefore lead to inflated expectations for C accrual.
**Agreed; we will reword this section to state that the fine fraction saturation is not a pertinent constraint on additional SOC storage capacity at depth and justify our focus on the maximal SOC accrual approach – citing Peoplau et al (2024, GCB).**
Lines 65-67: Perhaps the shortcomings of simple models could be briefly highlighted here, and at least one example of such a model provided in the text.
**We will cite Schimel 2023 to briefly mention the limitations of simple models (linear models dependent on C inputs as opposed to microbial models). We will also note in the Discussion that microbial models are not calibrated and remain an active field of research.**

Line 78: Can the model be introduced better here? Is it named? Why is this model appropriate for this use, etc?
**We will better introduce the model here as mentioned in the response to General Comment 3.**

Lines 79-81: Is this sentence supposed to mention a time horizon? Because couldn't the target levels be reached with different input rates, just on different time scales? But see comment below for line 217.
**This time horizon has been chosen because of national and international policies to reach C neutrality by 2050. This point will be mentioned in the revised version.**

Line 163: Yet, this method does not necessarily identify the maximum possible SOC stocks, it is constrained by several factors including input rates (if they increased massively, higher SOC levels could be reached). It is actually the maximum \*observed\* SOC stock, not necessarily the maximum possible. This really should be clarified here and throughout.
**We will clarify that the maximum stocks used as targets are theoretical maximum SOC stocks based on the 75th percentile of the SOC data observed within the region of study.**
Line 165: The Hassink method is not meant to estimate maximum total SOC stock, only the fine fraction. This and the previous comment point to the real need to clarify these concepts throughout the paper.
**We will remove the Hassink method from the main text.**
Line 168: I think using the term "data-driven approach" is misleading as the Hassink equation is data-driven. The distinction is more that one focuses on the fine fraction while the other does not, and that one was aiming to understand saturation (i.e. sought out particular soils with high fine fraction C contents) while the other is not (i.e., it is looking at maximum observed total C for a given dataset).
**We agree and will be more specific in the new version; the removal of the Hassink approach from the main text will also clear up this misunderstanding.**
Lines 169-174: Given the authors point out that improvements to the original Hassink equation have been suggested, what is the rationale for using the original Hassink equation? It is not clear.
**This section will be removed. To answer the Reviewer, we used the original equation because it tends to provide lower estimates compared to Feng and Georgiou. A lower estimate might limit the overestimation in how much carbon can be realistically added to soils within decadal timescales, especially at depth.**
Lines 178-179: These choices of percent POM and MAOM seem extremely arbitrary and overly precise. Why not use ranges? Please explain the rationale (even if briefly) rather than just referring to the papers.
**This section will be removed from the main text. In the appendix, the Hassink curve will be presented as a saturation in the fine fraction, and these estimates of POM and MAOM percentages will be removed.**
Lines 198-191: It is good that the authors highlight that the choice of boundary line affects the conclusions, but it is unclear why they chose 75th as the focus and 88th as the highest quantile when higher quantiles are commonly used in the literature (e.g. 95th; Georgiou et al. 2022).
**These percentile values are constrained by the logarithmic regression fit method, which first gives the 50th percentile, then the 75th, then the 88.5th, then the 94th. Other studies use the 95th percentile, but this was not applicable in our case due to the size of our depth-resolved dataset. Given that C inputs necessary to maintain SOC stocks at the 88th percentile at the steady state are excessively high, we consider it to be an acceptable higher quantile. We will make this more explicit in the text.**

Line 207: This is the first mention in the main text of the AMG model. Please define and explain what it is, ideally earlier in the manuscript. It is also important to describe how the model represents SOC pools of different character (including saturation behavior, temperature sensitivity) given the relevance of MAOM and POM to the overall interpretation.

**We will modify this section as detailed in our response to General Comment 3.**

Line 217: I find this to be very unclear and perhaps arbitrary. "... the vertical repartition of annual inputs needed to reach and maintain this target stock" by when? The time scale chosen will determine this level of inputs needed to reach the target (I suppose the point is that it is not overshooting the target, but that gets to whether the model and the maximum SOC potentials are realistic which is an issue in and of itself). Perhaps this is just due to wording, and it would be clearer to use only "maintain the target stock at steady state according to the model" or similar. Adding some of the detail that is now in the Appendix would also potentially help to clarify this. Overall though, this seems to be more of a result in itself; would it make more sense to choose realistic levels of inputs (or low-high ranges) and see how long it would take to reach the maximum C levels at steady state?

**We will modify this section to better explain our C input scenarios, as detailed in our response to General Comment 2, and make it clearer that we are interested in what can realistically be added to soil within 25 years.**

Line 229: The temperature aspect seems tacked on, yet I agree that it is an important issue for planning climate mitigation efforts. It could be more effectively introduced and discussed throughout the manuscript, including how the model handles temperature effects on SOC dynamics.

**We will mention in the introduction the effects of climate change (temperature, but also humidity) on SOC dynamics, and that different soil types will respond differently to the variations in environmental conditions. The new description of the model will also mention the consideration of temperature (General Comment 3).**

Line 249: Please clarify what is meant by "non-spatialized" (I got lost here in the explanation)

**We will clarify this important aspect of SOC stock mapping as suggested here: Due to the high spatial variability of soil characteristics, each mapping zone contains several soil types that cannot be explicitly delimited on the map at this spatial resolution. Therefore, each point within a given zone has a probability of belonging to one of several soil types (e.g: 70% chance of being a Eutric Cambisol, 30% chance of being a Stagnosol). The total SOC stock for this zone is given by the weighted mean of the SOC stocks (70 % of the SOC stock for Eutric Cambisols and 30 % of the SOC stock for Stagnosols). The standard deviation of the total SOC stock is given by the weighted standard deviations of the SOC stocks. The local uncertainty corresponds to expected local variations in the zone if the different soil types have contrasted SOC stocks. We visualize this local uncertainty by mapping the contrasts in SOC stocks within each zone in Appendix Figure 2.**

Line 363: In line with comments above, I do not think that use of the Hassink equation is "typical" as it has been surpassed by the boundary approaches mentioned in lines 169-174.

**This will be removed.**

Lines 400-403: I do not think it makes sense to directly compare the Chen et al. (2019) approach and the Georgiou et al. (2022) approach as they are very different in assumption and scope. Georgiou is assessing capacity of the fine fraction (i.e., saturation approach) while Chen is assessing ecosystem C level (i.e. potentially saturation-agnostic if saturation is not driving maximum total C levels). This gets at the general need to be clearer about the differences in these approaches throughout the paper, though I appreciate that this is done to some extent in lines 377-387.

**We agree that the two approaches explore different mechanisms. However, neither Chen et al (2019) nor Georgiou et al (2022) provide an explanation for their choice of percentile value. We believe our point still stands that the choice of percentile values remains empirical and lacks standardized rules based on the scale and scope of the study, and on the size and variability of the dataset.**

Paragraph beginning on line 449: I think this paragraph highlights the need to put these results into context of the input rates that were chosen, and what is realistic (see below). In theory, the determined maxima could be reached with very high input rates, and given that this paper does not include such scenarios it does not directly show that it is not achievable. But it does suggest so when one takes into account the input rates that would be required to do so, compared to those that were assumed in this scenario. This could be better explained here.

**See our new input scenarios in response to General Comment 2. We will modify our presentation of the input scenarios as detailed in response to General Comment 2, and we will discuss how realistic these inputs are in section 4.1 of the Discussion (see response to General Comment 4).**

Lines 458-469: This discussion of potential C inputs rates is not very thorough. Examples of input rates occurring in single studies is probably not representative of whether these can be achieved everywhere, especially when socioeconomic factors that limit changes to land management are considered. This part of the discussion should clearly acknowledge the difficulties in universally increasing C inputs to various lands, and ideally give ranges from the literature that demonstrate these realities. Further, as depth is such a key piece of this paper, this discussion should also touch on what is realistic in terms of inputs at depth (i.e. amount and vertical distribution), and whether there is evidence it can be achieved in different systems.

**We agree and will add a paragraph to address this in the discussion, section 4.2 (see response to General Comment 4)**

Line 690 (Appendix 1): I think that a figure which describes the model structure could be helpful.

**We added a better representation of the model structure in the new proposed Figure 2.**

Lines 691-699: Text is largely repeated from lines 200-209.

**We will correct repetitions in the text.**

Appendix equations: the terms in the equations need to be defined somewhere.

**This will be added.**

Lines 740-753: This information is key to understanding the logic and should be included in the main text. Yet, I still find this a bit confusing and am not clear on how the input rates presented in the main text were determined (how do they depend on the maximum potential SOC storage and time to reach it?).

**This will be added to our description of the model in the main text. For the C input scenarios, we will introduce them as in our response to General Comment 2.**

Line 750: The assumption that there is no vertical redistribution of C inputs, especially downward, is very unrealistic. The implications of this assumption for the results at depth should be described in the main text.

**We will add our justification for why not accounting for vertical transfer of C inputs at decadal timescales is acceptable in the main text, as detailed in our response to General Comment 3.**

Technical Corrections

Lines 169 and 214 (also Fig. 2 and elsewhere): I suggest an alternate term for "stationary", maybe "constant", "current", "baseline", "steady-state", or "equilibrium" depending on what is most correct here. **We will use the term steady-state.**

Line 391: change "ie." to "i.e.," **We will correct this.**

**References listed in our response that will be added to the new version of the paper:**

**Martin, M. P., Dimassi, B., Román Dobarco, M., Guenet, B., Arrouays, D., Angers, D. A., ... & Pellerin, S. (2021). Feasibility of the 4 per 1000 aspirational target for soil carbon: A case study for France. Global Change Biology, 27(11), 2458-2477.**

**Poeplau, C., Dechow, R., Begill, N., & Don, A. (2024). Towards an ecosystem capacity to stabilise organic carbon in soils. Global Change Biology, 30(8), e17453.**

**Schimel, J. (2023). Modeling ecosystem-scale carbon dynamics in soil: the microbial dimension. Soil Biology and Biochemistry, 178, 108948.**

**Sierra, C. A., Ahrens, B., Bolinder, M. A., Braakhekke, M. C., von Fromm, S., Kätterer, T., ... & Wang, G. (2024). Carbon sequestration in the subsoil and the time required to stabilize carbon for climate change mitigation. Global Change Biology, 30(1), e17153.**

---

## Author Comment (AC2)

This manuscript reports a study that simulated organic C accrual over 25 years in eight soils with contrasting properties of North-eastern France. The authors compared the Hassink equation and a novel data-driven approach to estimate soil organic C stocks and maximum soil organic C additional storage capacity. They found that the Hassink approach leads to unrealistic estimates and that the simulated soil organic C accrual over 25 years was five times lower than the maximum storage capacity.

The study has interest, and to the best of my knowledge, well conducted, although some assumptions need to be validated or better justified (see below). Also to the best of my knowledge, the results are well discussed and the conclusion are well supported. However, the study needs to be better introduced and some aspects of the approach need clarification.

**We thank you for your positive reception of the manuscript, and will try to answer your specific questions below:**

My specific comments:

l. 46. This sentence is unclear. What do you mean by "the upper percentiles of the total carbon content in a large dataset"? **The approach will be made clearer in our new figure 2 shown below. The theoretical maximum SOC stock corresponds to the 75th percentile curve of the carbon content dataset.**

[Figure]

**(a)** C(z)

Depth

+    Measured SOC
(per soil type and land use)

—    Mean initial SOC stock
(50th percentile)

- - -    Theoretical maximum SOC stock
(75th percentile)

▭    Theoretical maximum SOC accrual

**(b)** C Inputs per year ($I_{X - X\ cm}$)

Depth

➤    Vertical repartition of C inputs

**(c)** $I_{0 - 10\ cm}$    $a_1$    $a_2$

Fast (MRT~28)   Slow (~790)    $CO_2$    $CO_2$

$I_{10 - 20\ cm}$    $a_1$    $a_2$

~30   ~770    $CO_2$    $CO_2$

$I_{20 - 30\ cm}$    $a_1$    $a_2$

~20   ~920    $CO_2$    $CO_2$

$I_{30 - 40\ cm}$    $a_1$    $a_2$

~1390    $CO_2$    $CO_2$

...

**Suggested New Figure 2 : Summary of our approach: (a) estimation of initial and theoretical maximum SOC stocks from the measured data; (b) estimation of vertical repartition of C input for the different scenarios considered; (c) Functioning of the depth-dependent three-pool model (fast-cycling pool, slow-cycling pool, inert pool). a = allocation factor ; MRT = Mean Residence Time. MRT values vary with depth as per Balesdent et al. (2018) and are corrected for temperature, humidity, pH, texture and CaCO₃ (see Methods).**

l. 50 The Hassink method needs to be introduced. **Following comments from Reviewer 2, we will remove the Hassink approach from the main text. This will allow us to refocus on our main objective: the determination of realistic SOC accrual targets attainable within 25 years.**

l. 50. It is unclear what data-driven approach the authors are referring to. This should be better introduced and explained. **This approach will be referred to as the theoretical maximum SOC accrual approach. It will be better illustrated in the new version using the proposed New Figure 2.**

l. 64. Additional to what? Please clarify. **SOC additional storage capacity refers to what can realistically be added to the current SOC stocks, in our case within 25 years.**

l. 75. The last sentence seems to be disconnected from the rest of the paragraph. **We apologize if the logic was difficult to follow. In the new version, we will improve our definition of our C input scenarios by linking them more explicitly to realistic ranges found in the literature (details in response to General Comment 2 from Reviewer 2). This should clarify this paragraph.**

l. 78. The model needs to be briefly introduced here. **We will provide a more thorough definition of the model in the introduction and Material and Methods section (illustrated by Figure 2, see above). Please also see our reply to General Comment 3 from Reviewer 2 for details.**

l. 81-84. This text is unclear. **See our reply to Comment l. 75 above.**

l. 107. What do the authors mean by data points? **Each soil profile considered has several points of measurements at various depths, allowing us to plot SOC as a function of depth in the different soil types and land uses (See Figure 2). These are referred to as data points.**

l. 109. Sampling was conducted between and 2019. Do different starting points affect modeling results? **We assume, since land use has not changed since 1830 in the region of study, that the soil profiles are at steady state, and that sampling from 2009 to 2019 will not impact modelling results. This aspect will be made clearer in the new version of the paper.**

l. 130. The use of this pedotransfer function to estimate bulk density would need to be validated with data from this study. **This has been done for the data points where bulk density measurements were available (we compared different existing pedotransfer functions and selected the one that best estimated the measured bulk density values). We will mention this in the new version of the paper.**

l. 178-180. These estimates need justification and validation. **This section, and mentions of POM and MAOM, will be removed in the new version of the paper.**

l. 307. "for reasons that will be detailed further in the Discussion section" can be removed. **This will be removed.**

---

## Author Response (AR2)

**Reply to Comments**

We thank both the reviewer and the topic editor for their positive reception of the revised manuscript, and detail our additional corrections below following reviewer #2's suggestions:

**Anonymous Referee #2**

I appreciate the effort the authors have put into editing the manuscript according to reviewer suggestions, and believe it to be significantly improved. I have only two minor comments:

Please provide a reference for the suggested realistic inputs in forests, lines 215-218.

The following reference has been added (cited at line 217 and line 218): Mao, Z., Derrien, D., Didion, M., Liski, J., Eglin, T., Nicolas, M., Jonard, M., Saint-André, L. (2019). Modeling soil organic carbon dynamics in temperate forests with Yasso07. Biogeosciences, 16(9), 1955-1973. https://doi.org/10.5194/bg-16-1955-2019

It remains unclear as to how the authors got from estimates of inputs to the values chosen for scenario 3 (line 212), especially for grasslands and croplands. Some of this is mentioned later on, but it would be helpful to clearly explain why these specific values were chosen in the methods section."

We added the following sentence to clarify how the specific input values were chosen for scenario 3: "Within these ranges, the specific realistic values for the region of study were chosen by matrix inversion of the theoretical maximum SOC stocks, which provide the annual inputs necessary for the model to reach but to not exceed the maximum SOC stocks in the long term." (line 223-225)